# Whole brain correlates of individual differences in skin conductance responses during discriminative fear conditioning to social cues

**Kevin Vinberg[1]\*, Jörgen Rosén[1,2], Granit Kastrati[1,3], Fredrik Ahs[1]**

[1]Department of Psychology and Social Work, Mid Sweden University, Östersund, Sweden; [2]Department of Psychology, Uppsala University, Uppsala, Sweden; [3]Department of Clinical Neuroscience, Karolinska Institutet, Stockholm, Sweden

**Abstract** Understanding the neural basis for individual differences in the skin conductance response (SCR) during discriminative fear conditioning may inform on our understanding of autonomic regulation in fear-related psychopathology. Previous region-of-interest (ROI) analyses have implicated the amygdala in regulating conditioned SCR, but whole brain analyses are lacking. This study examined correlations between individual differences in SCR during discriminative fear conditioning to social stimuli and neural activity throughout the brain, by using data from a large functional magnetic resonance imaging study of twins (*N* = 285 individuals). Results show that conditioned SCR correlates with activity in the dorsal anterior cingulate cortex/anterior midcingulate cortex, anterior insula, bilateral temporoparietal junction, right frontal operculum, bilateral dorsal premotor cortex, right superior parietal lobe, and midbrain. A ROI analysis additionally showed a positive correlation between amygdala activity and conditioned SCR in line with previous reports. We suggest that the observed whole brain correlates of SCR belong to a large-scale midcingulo-insular network related to salience detection and autonomic-interoceptive processing. Altered activity within this network may underlie individual differences in conditioned SCR and autonomic aspects of psychopathology.

**\*For correspondence:** kevinvinberg@gmail.com

**Competing interest:** The authors declare that no competing interests exist.

## Editor's evaluation

Vinberg et al. provide a conceptual replication on individual differences in conditioned skin conductance response during fear acquisition training and BOLD fMRI in a large sample (*N* = 285) of healthy individuals (mono- and dizygotic twins). The authors report results that are in line with previous work and new results from a whole-brain analysis and suggest unique and shared contributions of individual brain regions.

## Introduction

Discriminative fear conditioning is one of the most common methods of studying fear learning in laboratory settings (*LeDoux, 2012*; *Davis and Whalen, 2001* and *Grillon, 2002*). It refers to the pairing of an initially neutral stimulus (CS+) with an aversive unconditioned stimulus (US), thereby imbuing the CS+ with the capacity to elicit increased autonomic responses relative to an unpaired neutral stimulus (CS-). Acquisition of conditioned fear forms an important part of theories of how anxiety and stress disorders develop from experiencing aversive events (*Mineka and Oehlberg, 2008*; *Bouton et al., 2001*; *Craske et al., 2014*). A more precise understanding of why some individuals are more

prone than others to developing anxiety and stress disorders may therefore be achieved by studying individual differences in fear conditioning. Ultimately, such knowledge could inform the development of more effective treatments for subgroups of patients whose responses to fear conditioning are related to the disorder (see e.g. *Insel, 2014*). Consequently, it has been argued that fear conditioning research should increasingly focus on understanding the sources of individual differences. One way that individuals vary during fear conditioning is in their autonomic responses (*Lonsdorf and Merz, 2017*).

The most commonly used autonomic measure of conditioned fear is the differential Skin Conductance Response (SCR; *Lonsdorf et al., 2017*), a label referring to the changes in skin conductance induced by sympathetic activation (*Dawson et al., 2007*; *Wallin, 1981*). A plethora of studies have investigated SCR in fear conditioning to determine its associations with psychiatric disorders. A meta-analysis of studies in patients with anxiety disorders found that SCR to the control cue (CS-) during the acquisition of conditioned fear was elevated in patients relative to controls (*Duits et al., 2015*). Duits et al. have proposed that increased SCR to the CS- was an effect of fear generalization from the threat cue to the control cue, or of reduced inhibition of threat responses to the control cue. Also, in individuals with schizophrenia, an increased SCR to the CS- has been reported across four independent fear conditioning studies (*Tuominen et al., 2022*). In OCD, however, a systematic review found associations to be less clear when considering SCR during the acquisition of conditioned fear and point to stronger OCD-related differences in SCR during the extinction phase of conditioned fear (*Cooper and Dunsmoor, 2021*). Taken together, these studies demonstrate increased SCR during fear conditioning to the control cue in patients with anxiety disorders and schizophrenia, relative to control participants, which gives credence to the notion that fear conditioning is an experimental model that can inform research on autonomic regulation in these disorders (*Lonsdorf and Merz, 2017*).

Previous studies of the neural correlates of individual differences in conditioned SCR, generally defined as the difference in average SCR score between CS+ and CS- presentations during acquisition (see *Lonsdorf and Merz, 2017*, for a discussion of definitions), have focused on either one or a few brain regions, using region of interest analyses. Many of these studies have found positive correlations with neural responses in the amygdala (*LaBar et al., 1998*; *Phelps et al., 2004*; *Dunsmoor et al., 2011*; *Petrovic et al., 2008*; *MacNamara et al., 2015*; *Marin et al., 2020*). Neuroimaging studies of within-subject variation in conditioned SCR have also generally found positive correlations to amygdala responses (*Cheng et al., 2003*; *Knight et al., 2005*; *Cheng et al., 2006*), although exceptions exist (*Sjouwerman et al., 2020*; *Savage et al., 2021*). The findings from studies that report a positive relationship between SCR and amygdala activity are in line with the general understanding of fear conditioning from animal models, where a neural circuitry centered on the amygdala is responsible for the acquisition of conditioned fear responses (*LeDoux, 2000*; *Davis, 2000*). They also complement those human lesion studies demonstrating either diminished or absent conditioned SCR following amygdala damage (*LaBar et al., 1995*; *Bechara et al., 1995*), although not all studies have found such an effect (*Ahs et al., 2010*; for a review see *Ojala and Bach, 2020*). Further, the involvement of the amygdala in human fear conditioning has been questioned based on the results of a meta-analysis of fMRI studies investigating fear conditioning (*Fullana et al., 2016*) and based on studies showing unexpected, increased amygdala responses to the CS- compared to the CS+ (see e.g. *Visser et al., 2021*). Such results could arise from distributed representations of the CS+ and CS- in the amygdala (*Bach et al., 2011*; *Reijmers et al., 2007*) or from a need for larger sample sizes to detect differential responses in the amygdala. Speaking to the latter idea, two independent studies, each including hundreds of participants, have recently reported increased CS+, relative to CS-, activation in the amygdala (*Kastrati et al., 2022*; *Wen et al., 2022*). Amygdala activation to the CS+ was primarily detected during the first trials of acquisition, whereas CS- activity was larger in the end of acquisition (*Wen et al., 2022*). The results of these two large and independent neuroimaging studies, together with the fairly consistent findings of correlated individual differences in conditioned SCR and amygdala activation (*LaBar et al., 1998*; *Phelps et al., 2004*; *Dunsmoor et al., 2011*; *Petrovic et al., 2008*; *MacNamara et al., 2015*; *Marin et al., 2020*), support the hypothesis that amygdala activation should be positively correlated with SCR.

A limiting factor of previous studies of the neural correlates of individual differences in conditioned SCR is that they generally have been based on small sample sizes ($N \leq 27$) and have reported results from region of interest analyses (ROIs). Notably, most previous studies have sample sizes that fall

**Table 1.** Previous studies examining the association between individual differences in skin conductance responses (SCR) and neural activation during fear conditioning.

| Study | Participants (n) | Analysis | Definition of individual SCR scores | Definition of neural activation | ROI(s) | Statistical threshold | Findings |
|---|---|---|---|---|---|---|---|
| *LaBar et al., 1998* | 5 | Correlation | CS+ minus CS- | No. of voxels in CS+ > CS- contrast | Amygdala, rostral and caudal ACC | $P_{Unc} < 0.001$ | Positive correlation in Amygdala |
| *Phelps et al., 2004* | 11 | Correlation | CS+ minus CS- | CS+ > CS- contrast | Amygdala, mid PFC | $P_{Unc} < 0.001$ | Positive correlation in Amygdala |
| *Dunsmoor et al., 2011* | 14 | Correlation | CS+ minus CS- | CS+ > CS- contrast | Whole brain, Amygdala | Whole brain: $P_{Unc} < 0.001$ then ROI: $P_{FWE} < 0.05$ | Positive correlation in left Amygdala |
| *Petrovic et al., 2008* | 27 | Correlation | Late(CS+ minus CS-) - Early(CS+ minus CS-) | CS+ > CS- contrast | Amygdala, Fusiform Gyrus and pain regions from *Peyron et al., 2000* | $P_{Unc} < 0.001$ | Positive correlation in Amygdala |
| *MacNamara et al., 2015* | 49 | Correlation | CS+ minus CS- | CS+ > CS- contrast | Amygdala, Insula, ACC, cerebellum (lobule 4–5), mPFC, precentral gyrus, STG | $P_{Unc} < 0.001$ then $P_{FWE} < 0.05$ using ClusterSim | Positive correlation in right Amygdala and left SMA. Positive correlation in left Amygdala using relaxed statistical threshold. |
| *Marin et al., 2020* | 60 | High vs. low SCR responders (drawn from larger sample of N = 109) | CS+ minus CS- | CS+ > CS- contrast | Amygdala, Insula, dACC, sgACC, vmPFC | Combined ROIs: $P_{Unc} < 0.001$; Single ROIs: $P_{FWE} < 0.05$ | High SCR responders > Low responders: Left Amygdala, left Insula and vmPFC |

Note: ACC, Anterior cingulate cortex; dACC, dorsal ACC; FWE, Family-wise error, mPFC, medial PFC; PFC, Prefrontal cortex; sgACC, subgenual ACC; Unc, Uncorrected; vmPFC, ventromedial PFC.

below the minimum guidelines for correlation analysis in fMRI research (*Yarkoni, 2009*; *Yarkoni and Braver, 2010*), which recommend a sample size of at least *N* = 40 for determination of inter-individual correlations. Studies that do not comply with this minimum recommendation are highly susceptible to type II errors (i.e. not detecting real effects, even within ROIs) as well as gross inflation of reported effect sizes (*Yarkoni, 2009*). Furthermore, some of the reported studies also use non-standard statistical procedures, such as reporting results from uncorrected whole brain analyses (*Petrovic et al., 2008*), or initially use an uncorrected whole brain analysis followed up by stringent FWE-correction within masks only targeting regions implicated in the prior uncorrected analysis (*Dunsmoor et al., 2011*), which raises the risk for type I error (see e.g. *Poldrack et al., 2017*) and could reduce the reliability of the results. Only two previous studies that investigated the association between individual differences in conditioned SCR and neural activity have met the minimum requirements for sample size suggested by *Yarkoni and Braver, 2010*: *MacNamara et al., 2015* (*N* = 49) and *Marin et al., 2020* (*N* = 60). However, both of these studies have still reported results based on uncorrected statistics in pre-defined ROIs (See *Table 1* for a comparison of previous studies investigating neural correlates of SCR).

The previous focus on ROIs excludes activations in many parts of the brain that are potentially important for explaining individual differences in conditioned SCR. Fear conditioning is known to activate a large set of cortical, subcortical, and brainstem areas other than the ROIs that have so far been investigated for their correlation with SCR (*Fullana et al., 2016*). In their meta-analysis, *Fullana et al., 2016* proposed that neural regions consistently activated during fear conditioning collectively constitute a large-scale neural network, centered on the dACC and anterior insula, that represents autonomic-interoceptive processing in response to conditioned stimuli (*Fullana et al., 2016*; based on findings by e.g. *Cameron, 2009*; *Craig, 2009*; *Critchley and Harrison, 2013*; *Medford and Critchley, 2010*). Based on this proposal, one would expect individual differences in autonomic conditioned responding, such as those measured by SCR (*Dawson et al., 2007*), to correlate with conditioning-related activity within this broader network.

Because previous studies only studied correlations to SCR in a handful of brain regions and in relatively small samples, the primary aim of this study was to investigate the whole brain correlations of individual differences in conditioned SCR by analyzing data from a large twin sample performing a fear conditioning task (*N* = 285 individuals). Similar to previous studies (e.g. *Phelps et al., 2004*; *Dunsmoor et al., 2011*; *MacNamara et al., 2015*), the present study used differential SCR as a between-subjects regressor of CS+ > CS- BOLD activation. Previous studies have not found whole-brain correlations surviving correction for multiple comparisons. In the current study, the sample size

of $N$ = 285 individuals provided sufficient statistical power to detect correlations of medium effect size and above ($r$ > .251) throughout the whole brain (see section 4.3.3 in the Materials and methods section for details), thereby substantially improving upon the power of previous whole brain analyses of smaller sample sizes. A family-wise error (FWE) corrected alpha level of $\alpha$ = .05 was used to constrain the risk of type I errors to an acceptable level (see e.g. *Poldrack et al., 2017*), which was not done in previous studies.

Based on findings from previous studies of individual differences in conditioned SCR (*LaBar et al., 1998*; *Phelps et al., 2004*; *Dunsmoor et al., 2011*; *Petrovic et al., 2008*; *MacNamara et al., 2015*; *Marin et al., 2020*), we hypothesized a positive correlation between SCR and amygdala activation. Again, our sample size yielded substantially improved power in comparison to previous studies while securing an alpha level of $\alpha$ = .05 (see section 4.3.3 in the Materials and methods section for details). Finally, a last aim of the present study was to determine whether areas whose activity explained significant individual variation in conditioned SCR did so independently of one other. A comparison of the relative contributions of different brain areas to conditioned SCR has the potential to elucidate separate neural pathways mediating individual differences in conditioned autonomic responding. Ultimately, such findings may aid the understanding of autonomic regulation in pathological fear and anxiety.

## Results
### SCR
There was no difference in SCR to CS+ and CS- during habituation ($M$ = 0.35, $SD$ = 0.60; $t(285)$ = 0.27; $p$ = 0.79). During acquisition, participants displayed significantly larger SCRs to the CS+ relative to the CS- ($t(284)$ = 23.28; $p$ < 0.001; $d$ = 1.38), indicating successful conditioning. An SCR difference score between the CS+ and the CS- was calculated for each participant ($M$ = 0.64, $SD$ = 0.47). The distribution of SCR difference scores revealed substantial individual differences (see Appendix 1). The average shock expectancy was greater to the CS+ ($M$ = 0.68, $SD$ = 0.26) than the CS- ($M$ = 0.09, $SD$ = 0.21; $t_{285}$=25.52; $p$ < 0.05).

### Brain responses
The effects of fear conditioning on fMRI responses during habituation and acquisition (CS+ > CS-) are described in Appendix 6. We found no differences in neural responses to the CS+ compared to the CS- during habituation. During acquisition, the pattern of activation to the CS+ relative to the CS- was very similar to the pattern reported in the meta-analysis by *Fullana et al., 2016* and included large parts of the striatum, the insula, midline areas of the cingulum, lateral temporal cortex, parietal cortex, and the supplementary motor areas. Of note, the whole brain analysis also revealed greater activation to the CS+ than to the CS- bilaterally in the amygdala.

### Correlation between SCR difference scores and brain responses during fear conditioning
#### Whole brain analysis
Fear conditioning-related brain responses (CS+ > CS-) were correlated to SCR difference scores in the dorsal anterior cingulate cortex/anterior midcingulate cortex, right anterior insula, right inferior frontal gyrus/frontal operculum, bilateral temporoparietal junction/superior temporal gyrus, right superior parietal lobe/postcentral gyrus, bilateral superior frontal gyri/dorsal premotor cortex, and a right-lateralized midbrain region in areas consistent with periaqueductal gray and reticular formation. For a summary of results, see *Figure 1* and *Table 2*. In order to ensure the reliability of our findings and facilitate comparison with prior studies, we performed the same analysis using average square root transformed raw value SCRs instead of Z transformed SCRs. We also performed the same analysis without any participant exclusion (see the Materials and methods section regarding participant exclusion). Both of these analyses resulted in a very similar pattern of correlations (see Appendix 3 and 4), showing that the results seem robust to different types of SCR normalization and to different choices regarding participant exclusion. In addition, we also repeated our analysis using an extended SCR response window as well as controlling for shock expectancy and genetic influence, again with similar results (see *Appendix 7—table 1*). Finally, as pointed out by a reviewer, the Ledalab software

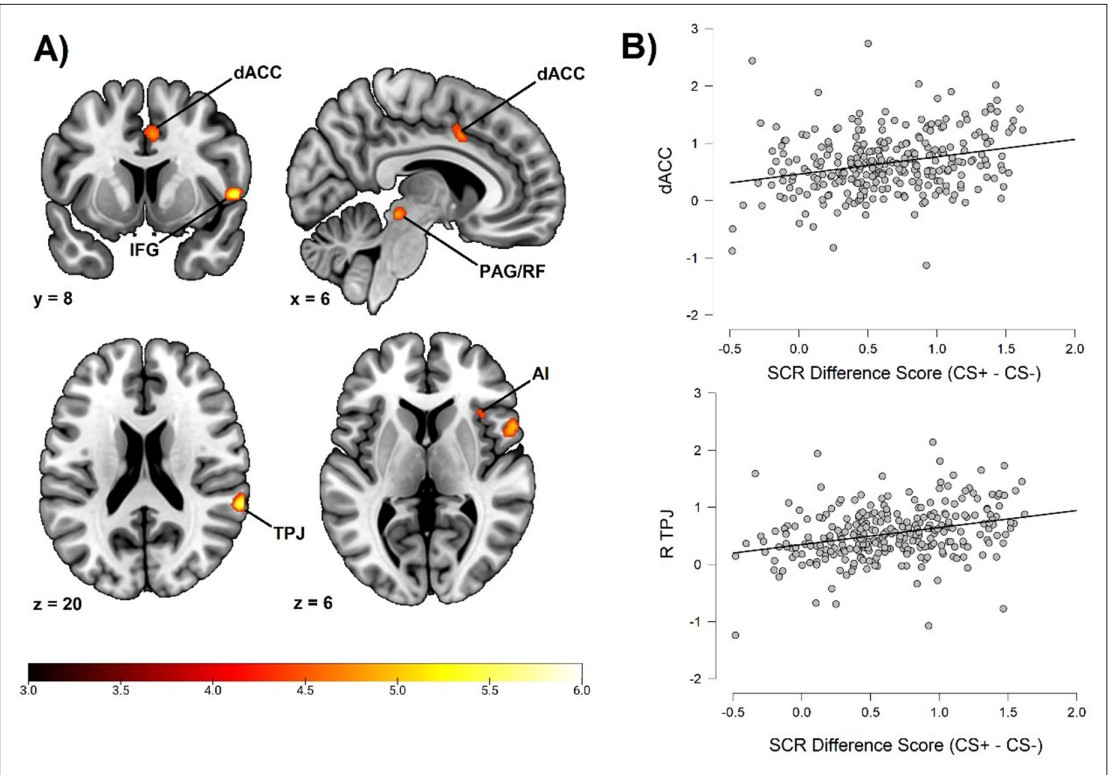

**Figure 1.** Correlation between individual differences in conditioned SCR and whole brain responses during fear conditioning, obtained using individual SCR scores (Z transformed average CS+ minus CS- SCR) as a second level, between-subjects regressor of the average CS+ > CS- BOLD activation in SPM12 (Wellcome Department of Cognitive Neurology, University College, London) software. The sample consisted of 285 participants who passed the following exclusion criteria: pregnancy, inability to lie still for a 1 hr duration, intolerance of tight confinements, ongoing psychological treatment, metal objects in the body (due to surgery, fragmentation, etc.), current alcohol or drug-related problems, use of psychotropic medications, unsuccessful recording of skin conductance responses, loss of brain imaging data due to excessive head movement, and participant failure to comply with task instruction regarding button press in at least 80% of trials. (**A**) Activation map of key implicated neural regions. Color-coded $t$ values ranged from $t = 3$ to $t = 6$. The statistical image was thresholded at $P < 0.05$ FWE-corrected and displayed on an anatomical brain template. (**B**) Scatter plots depicting correlation between SCR difference scores and eigenvariates from significant whole brain clusters in the dorsal anterior cingulate cortex (upper panel) and the temporoparietal junction (lower panel). $R$ = right dACC = dorsal anterior cingulate cortex. TPJ = temporoparietal junction. IFG = inferior frontal gyrus. PAG/RF = periaqueductal gray/reticular formation. AI = anterior insula.

The online version of this article includes the following source data for figure 1:

**Source data 1.** Variable data used to produce *Figure 1B*, *Figure 2B* and statistical analyses reported in the section 'Relative contribution of neurofunctional correlates to individual differences in SCR', as well as *Appendix 1—figure 1*, *Appendix 2—figure 1* and *Appendix 5—table 1*.

package (v 3.4.9; *Benedek and Kaernbach, 2010*) presently used for SCR scoring has been shown to yield no better and sometimes worse results than standard peak scoring (*Bach, 2014*). For this reason, we also repeated our whole brain analysis using the PsPM software package (v 5.1.1) (*Bach and Friston, 2013*). This analysis once again implicated the same set of regions. See Appendix 7 for results and Appendix 8 for additional information on how this analysis was conducted (complementary analysis).

## Correlation between individual differences in conditioned SCR and amygdala activation

To test the hypothesis of a correlation between individual differences in conditioned SCR and amygdala activation, we performed an ROI analysis. This analysis demonstrated significant correlations bilaterally in the amygdala (right peak MNI coordinates: 20, –2, –14; cluster size = 38 voxels; $t = 3.68$; left peak MNI coordinates: –22, –2, –16; cluster size = 8 voxels; $t = 3.08$). See *Figure 2*.

**Table 2.** Whole brain correlation to conditioned SCR.

| Anatomical region | Hemisphere | Voxels | t | MNI Coordinates x | y | z |
|---|---|---|---|---|---|---|
| Dorsal Anterior Cingulate Cortex/Anterior Midcingulate Cortex | N/A | 50 | 4.79 | 6 | 8 | 40 |
| Anterior Insula | Right | 20 | 4.65 | 36 | 20 | 6 |
| Inferior Frontal Gyrus/Frontal Operculum | Right | 138 | 5.80 | 56 | 10 | 2 |
| Temporoparietal Junction/Superior Temporal Gyrus | Right | 81 | 5.66 | 64 | –40 | 20 |
| Superior Frontal Gyrus/Dorsal Premotor Cortex | Right | 44 | 5.21 | 18 | 0 | 68 |
| Midbrain | Right | 59 | 5.22 | 10 | –30 | –12 |
| Superior Parietal Lobe/Postcentral Gyrus | Right | 3 | 4.52 | 22 | –46 | 70 |
| Superior Frontal Gyrus/Dorsal Premotor Cortex | Left | 2 | 4.46 | –14 | -2 | 72 |
| Temporoparietal Junction/Superior Temporal Gyrus | Left | 1 | 4.37 | –62 | –36 | 22 |

Note. MNI coordinates and t values represent significant peak voxels of each cluster. Statistical significance was calculated using t tests implemented within the SPM software with an FWE corrected alpha level of α = .05.

## Relative contribution of neurofunctional correlates to individual differences in SCR

In order to examine the independent and/or shared contributions of neural responses to SCR, we extracted eigenvariates of contrast values from all significant clusters in the whole brain analysis.

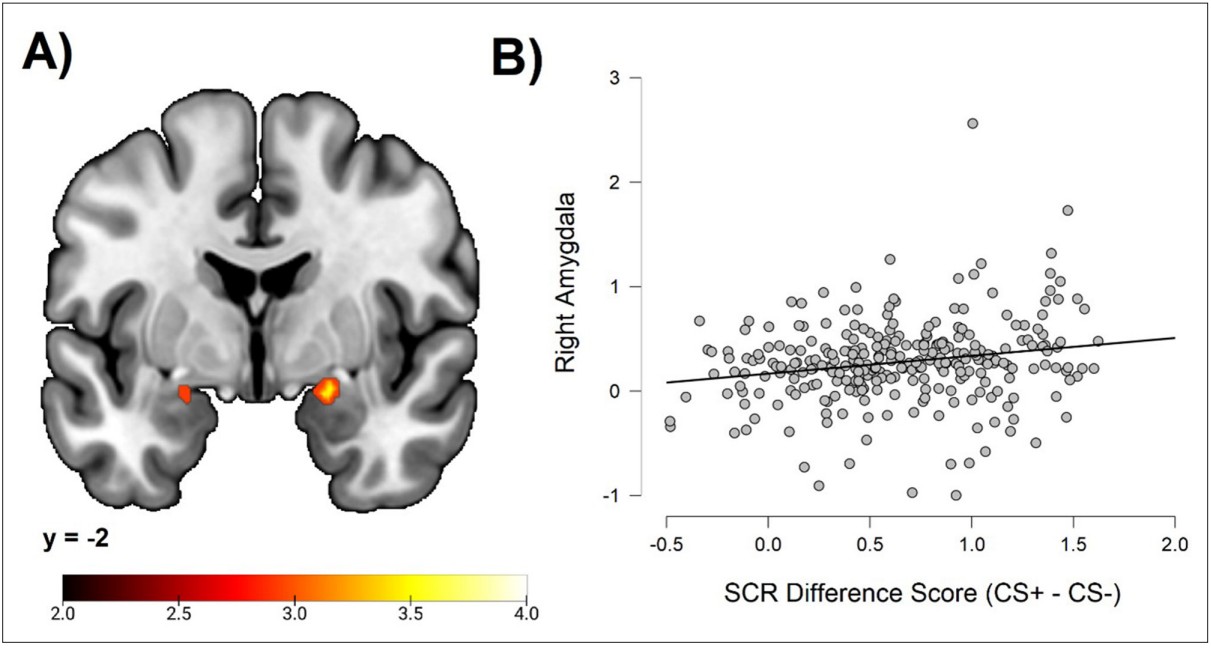

**Figure 2.** Correlations between individual differences in conditioned SCR and amygdala activation, obtained using individual SCR scores (Z transformed average CS+ minus CS- SCR) as a second level between-subjects regressor of the average CS+ > CS- BOLD activation in SPM12 (Wellcome Department of Cognitive Neurology, University College, London) software. The sample consisted of 285 participants who passed the following exclusion criteria: pregnancy, inability to lie still for a 1 hr duration, intolerance of tight confinements, ongoing psychological treatment, metal objects in the body (due to surgery, fragmentation, etc.), current alcohol or drug-related problems, use of psychotropic medications, unsuccessful recording of skin conductance responses, loss of brain imaging data due to excessive head movement, and participant failure to comply with task instruction regarding button press in at least 80% of trials. (**A**) Activation map depicting significant activation on coronal section at MNI Y-coordinate = –2. Color-coded t values range from t = 2.0 to t = 4.0. The statistical image was thresholded at p < 0.05 FWE-corrected. (**B**) Scatter plot depicting correlation between SCR difference scores and eigenvariates from the significant right amygdala cluster within the amygdala ROI. For source data to (**B**), see *Figure 1—source data 1*.

Eigenvariates were then entered as regressors in a hierarchical regression analysis. Together, regional eigenvariates demonstrated a significant correlation with conditioned SCR with a moderate effect size ($F(1, 284) = 4.82$; $r = .37$; $r^2 = .14$; $p < 0.001$). No region contributed unique, statistically significant variance (see Appendix 5). Source data for all hierarchical regression analyses can be found in *Figure 1—source data 1*.

Individual differences in SCR difference scores could be associated with individual differences in SCR to both the CS+ and the CS-. Therefore, we wanted to test whether SCR to the CS+, and not the CS-, was the reason for the observed correlation between SCR difference scores and eigenvariates. To this end, we correlated the extracted eigenvariates from regions that were correlated to SCR difference scores in the whole brain analysis with average raw value SCRs to the CS+ and CS-, separately. Results demonstrated significant correlations between all regional BOLD eigenvariates and CS+ SCR ($p$-values $\leq 0.001$), except for right superior parietal lobe ($p = 0.002$), left superior frontal gyrus ($p = 0.002$), right amygdala ($p = 0.006$), and left amygdala ($p = 0.029$), where correlations did not survive the Bonferroni-corrected threshold ($\alpha = 0.00151$; see the Materials and methods section for details regarding the Bonferroni-correction). However, no correlations between SCR to the CS- and extracted eigenvariates were significant ($p$-values $> 0.00151$), and all extracted values were significantly more correlated to CS+ SCR than CS- SCR ($p$-values $< 0.00151$) except for the right ($p = 0.018$) and left amygdala ($p = 0.011$). This indicated that neural correlations to differential SCR were mainly explained by increased responding to the CS+. Source data for all correlation analyses involving the CS+ and CS- separately can be found in *Figure 1—source data 1*.

## Discussion

Individual differences in SCR during discriminative fear conditioning are common (*Lonsdorf and Merz, 2017*) and have been associated with psychopathology (*Duits et al., 2015*; *Nees et al., 2015*; *Lonsdorf and Merz, 2017*). In this study, we examined the whole brain correlates of conditioned SCR in a large sample of twins ($N = 285$) during discriminative fear conditioning with concomitant SCR and fMRI recordings. As expected, we found a correlation between individual differences in conditioned SCR and amygdala activity in line with previous reports (*LaBar et al., 1998*; *Phelps et al., 2004*; *Petrovic et al., 2008*; *Dunsmoor et al., 2011*; *MacNamara et al., 2015*; *Marin et al., 2020*). Our analysis also implicated the dACC and insula, two regions which have previously been found to show increased responses in high vs. low conditioners (*Marin et al., 2020*). Importantly, we also identified correlations in novel regions including the bilateral temporoparietal junction/superior temporal gyri, right frontal operculum, bilateral dorsal premotor cortex, right superior parietal lobe, and midbrain. All correlations between brain responses and conditioned SCR were positive, meaning that individuals who respond more strongly to the CS+ relative to the CS- on the physiological level (SCR) also showed greater neural activation to the CS+ relative to the CS-. Furthermore, all regional activations demonstrated a stronger correlation with SCR to the CS+ alone compared to the CS- alone, although a few correlations did not survive Bonferroni correction. This indicated that neural activity was primarily associated with heightened, conditioned SCR to the CS+ (i.e. as opposed to inhibited SCR to the CS-).

An important research question is whether the neural network associated with individual differences in conditioned SCR is embedded in the network of regions that is generally activated during fear conditioning. The whole brain correlates of SCR found in the present study belong to the set of regions consistently activated during human neuroimaging studies of fear conditioning (*Fullana et al., 2016*). This indicates that these regions not only respond to the CS+ relative to the CS- in general, but that the magnitude of this activation also co-varies with individual differences in the magnitude of conditioned responding indexed by SCR. This is consistent with the proposal by *Fullana et al., 2016*; based on findings by e.g. *Cameron, 2009*; *Craig, 2009*; *Critchley and Harrison, 2013*; *Medford and Critchley, 2010* that these regions, especially the dACC and anterior insula, are part of a large-scale neural network regulating autonomic responding (SCR). Reasonably, increased autonomic activation would correlate with larger responses in neural regions regulating autonomic processing. To better understand the potentially unique contributions of different brain regions to SCR, we performed a hierarchical regression analysis. Results suggested that cortical areas together with midbrain regions contributed to individual differences in conditioned SCR. No unique contribution from any of the regions could be proven. These results are consistent with the idea that the regions identified in the whole brain analysis form part of one functional network, rather than separate independent

networks. Indeed, the functional connectivity and network-structure between the regions reported in the present study have been examined by several research groups previously (see e.g. *Uddin et al., 2019*, for a review). These research groups have proposed a more general function for this network, beyond autonomic-interoceptive processing and autonomic regulation, in detecting and preparing responses to salient events across homeostatic, affective, and cognitive domains (see e.g. *Seeley et al., 2007*; *Menon and Uddin, 2010*; *Uddin, 2015*; *Menon, 2015*; *Uddin et al., 2019*). *Uddin et al., 2019* refer to this network as the 'midcingulo-insular network' to reflect the anatomy of the network, rather than using functional labels that can be dependent on context (e.g. 'salience network', *Menon, 2015*; 'ventral attention network', *Corbetta et al., 2008*). Within this network, the dACC/aMCC (see *Vogt, 2016*, for a discussion regarding the naming of this region) and anterior insula, constitute the major input-output nodes (*Uddin et al., 2019*; *Menon, 2015*). While, the insula is thought to integrate cognitive, affective, interoceptive, and homeostatic information, the dACC is believed to represent this summarized information in order to determine autonomic, behavioral, and cognitive responding (*Menon, 2015*; *Medford and Critchley, 2010*). Efferent autonomic output from the dACC is proposed to be mediated by the periaqueductal gray (*Menon, 2015*), which was another region identified in our whole brain analysis (labeling consistent with a previous definition of the peri-aqueductal gray, see *Linnman et al., 2012*). Notably, the midcingulo-insular network also includes the bilateral temporoparietal junction/superior temporal gyri (*Uddin et al., 2019*; see e.g. *Yeo et al., 2011*; *Seeley et al., 2007*; *Bzdok et al., 2013*), whose activity we found to be correlated with SCR. The right frontal operculum, which was another region we found to be positively correlated to SCR, is part of what has previously been called the 'cingulo-opercular network' or the 'ventral attention network' (*Uddin et al., 2019*; see e.g. *Corbetta et al., 2008*). More specifically, the right lateralized temporoparietal junction and frontal operculum appear to be recruited particularly during exogenous salience detection (*Uddin et al., 2019*), which fits well with the stimulus-driven salience detection in the context of fear conditioning. Finally, the amygdala has also been proposed to constitute a major subcortical node within this network (*Menon, 2015*; *Uddin et al., 2019*). Thus, most of the regions whose activity we found to correlate with SCR belong to this functional network, consistent with its role in regulating autonomic responses, the only exception being the bilateral dorsal premotor cortex. However, the premotor cortex is known to be a major cortical elicitor of the SCR (*Dawson et al., 2007*; *Boucsein, 2012*), making it a plausible link between midcingulo-insular activity and peripheral skin conductance. Consequently, based on our findings and on previous evidence-based theory, we propose that individual differences in conditioned SCR may originate from activity within the midcingulo-insular network (*Uddin et al., 2019*) in conjunction with the dorsal premotor cortex.

It has been suggested that understanding sources of individual differences in fear conditioning may uncover individual risk and resilience factors with respect to fear and anxiety that may ultimately aid the understanding and treatment of fear-related psychopathology (*Lonsdorf and Merz, 2017*). Our proposal that individual differences in conditioned SCR co-vary with activity in a large-scale neural network substantiating autonomic-interoceptive processing and salience detection may high-light such a source. Indeed, overactivity within the midcingulo-insular network has previously been suggested to underlie pathological anxiety (*Menon, 2011*; *Menon, 2015*), as patients with anxiety disorders show altered functional connectivity within this network (*Peterson et al., 2014*) as well as hyperactivity within the anterior insula (*Paulus and Stein, 2006*; *Stein et al., 2007*) and amygdala (*Etkin and Wager, 2007*), which are important nodes of the network. Based on these findings, it has been suggested that anxiety disorders and neuroticism may result from excessive processing of emotion-related salient cues (*Menon, 2011*) and/or alterations in autonomic interoceptive-autonomic processing (e.g. *Paulus and Stein, 2006*; *Medford and Critchley, 2010*). As anxiety disorders have also been associated with altered discriminative fear learning (*Nees et al., 2015*), and as our results indicate that individual differences in discriminative fear learning covary with midcingulo-insular activity, our results are consistent with this anxiety model. Specifically, in reviewing the relationship between discriminative fear learning and anxiety disorders, *Nees et al., 2015* note that while discriminative fear learning does not appear to be impaired across all anxiety disorders and across all stimuli types (see also *Duits et al., 2015*, for a meta-analysis confirming this observation), studies comparing anxiety disorder patients to controls with regard to disorder-specific stimuli have found increased discriminatory fear learning in patients (in specific phobia, see *Schweckendiek et al., 2011*; in social phobia, see *Lissek et al., 2008*; in PTSD, see *Wessa and Flor, 2007*), suggesting that

discriminatory fear learning may be a mechanism underlying these disorders. The present finding, that individual differences in discriminatory fear learning covary with differences in midcingulo-insular activity, suggests that alterations in such neural activity may also be contributing to, or resulting from, anxiety, consistent with the anxiety model proposed by *Menon, 2011*. However, it should be noted that the present study only used SCR as an outcome measure of discriminative fear learning while the previously cited studies considered other outcome measures as well (e.g. fear potentiated startle in *Lissek et al., 2008*; subjective ratings of valence and arousal in *Wessa and Flor, 2007*), thus somewhat limiting the generalizability of our findings (for further details regarding methodological differences and similarities we refer to the review by *Nees et al., 2015*). We recommend that future studies continue exploring the midcingulo-insular network and its relationship to fear and anxiety disorders.

One limitation of the present study is the use of social stimuli as CSs. While the general CS+> CS- BOLD contrast analyzed in the present study largely demonstrated a pattern of activation typical to fear conditioning (*Fullana et al., 2016*; see Appendix 6), the potential influence of social threat processing cannot be entirely ruled out. Therefore, we recommend reproducing our results using non-social stimuli as CSs. Another limitation is the use of a twin sample, which may affect both the generalizability and validity of our findings. Regarding generalizability, small to moderate differences between twins and the normal population have been reported for some measures of fear and anxiety (*Munn-Chernoff et al., 2013*; *Kendler et al., 1995*) but not others (*Pulkkinen et al., 2003*). To what extent twins diverge from the normal population with regards, specifically, to neural and physiological responding during fear conditioning is largely unknown, and, while we know of no reason to assume a significant difference between the two populations, ideally our results should be extended to a random sample from the normal population. Furthermore, regarding the validity of our findings, there is evidence demonstrating moderate heritability of SCR (*Hettema et al., 2003*) and BOLD-responses (*Kastrati et al., 2022*) during fear conditioning, meaning that twin pairs may be more similar to each other than expected by chance and therefore making data points less independent than normally assumed. To what extent this affects the reported associations to neural activity is difficult to determine. In order to test for any biases, we repeated our analysis using only single-twins, consequently cutting our sample size in half (see Appendix 7). All implicated regions evidenced correlation coefficient strengths similar, or only slightly lower, than those resulting from use of the full sample. This indicated that our results from the full sample were also applicable to unrelated individuals (see Appendix 7). In summary, while the use of a twin sample is a limitation of the present study, we contend that there is no reason to conclude that it invalidates the reported results. Finally, it should be noted that only a few studies to date have examined the longitudinal (test-retest) reliability of individual differences in discriminatory fear learning measured by SCR (*Cooper et al., 2022*, Preprint; *Klingelhöfer-Jens et al., 2022*, Preprint). While one study found evidence of fair within-person stability in 51 participants across a 9-day period (*Cooper et al., 2022*, Preprint), another study found evidence of poor individual-level reliability in 120 participants across a 6-month period (*Klingelhöfer-Jens et al., 2022*, Preprint). Similarly, previous studies examining the individual-level reliability of fMRI BOLD responding during fear conditioning have reported low to moderate reliability (*Ridderbusch et al., 2021*; *Klingelhöfer-Jens et al., 2022*, Preprint). Taken together, this means that it is currently unclear to what extent our results reflect stable individual traits, as opposed to participants' particular states at the time of measurement, which limits the interpretation of our findings. In line with *Klingelhöfer-Jens et al., 2022* (Preprint) we encourage future research on individual differences in fear conditioning to explore new ways of improving the reliability of measurement.

In summary, the present study is the largest study to date on the neural correlates of autonomic fear conditioning and identified several novel areas whose activations predict individual differences in conditioned SCR. Previous findings from smaller studies could also be confirmed. Our results are consistent with the activation of a large-scale midcingulo-insular network substantiating autonomic-interoceptive processing and salience detection. We propose that altered activity within this network underlies individual differences in conditioned SCR and possibly autonomic regulation in pathological fear and anxiety. Future research should continue investigating this network as well as its possible relationship to fear and anxiety. Ultimately, such efforts may uncover the mechanisms of fear-related psychopathology and aid in its treatment.

## Materials and methods

### Participants

This study was part of a twin study of genetic influences on fear-related brain functions. Results describing genetic influences on SCR and fMRI responses during fear conditioning will be reported elsewhere. A total of 311 adult monozygotic ($N$ = 138) and dizygotic ($N$ = 147) twin volunteers were recruited from the Swedish Twin Registry (Svenska Tvillingregistret). Six participants were excluded before data collection because they were unable to undergo magnetic resonance imaging. After data collection, another twenty participants were excluded from analysis due to one or several of the following reasons: unsuccessful recording of skin conductance responses (2 participants); loss of brain imaging data due to excessive head-movement (5 participants); failure to indicate shock expectancy with button press in at least 80% of trials (11 participants; see section 4.3.1); use of psychotropic medication (7 participants). Thus, 285 participants (female = 167, mean age = 33.92 years, age range = 20–58 years, $SD$ = 10.11 years) were included in the analyses. All of these 285 participants (138 monozygotic and 147 dizygotic twins) passed the following exclusion criteria: pregnancy, inability to lie still for a 1 hr duration, intolerance of tight confinements, ongoing psychological treatment, metal objects in the body (due to surgery, fragmentation, etc.), ongoing substance abuse, or use of psychotropic medications. Non-psychotropic medication was not an exclusion criterion. However, in order to ensure the reliability of our findings, we also performed an additional supplementary analysis wherein all participants with fMRI and SCR data were included ($N$ = 303; see Appendix 4). Participants receiving psychological treatment remained excluded as treatment could affect brain responses to emotional stimuli. Although this may be less problematic for the current analysis of brain correlates of SCR, it could be a problem when analyzing correlations between members of a twin pair if one individual receives treatment and one does not (these results will be reported elsewhere). Participants provided written informed consent in accordance with guidelines from the Regional Ethical Review Board in Uppsala and received SEK 1000 as reimbursement for their participation. The study protocol was approved by the Regional Ethical Review Board in Uppsala (Dnr 2016/171).

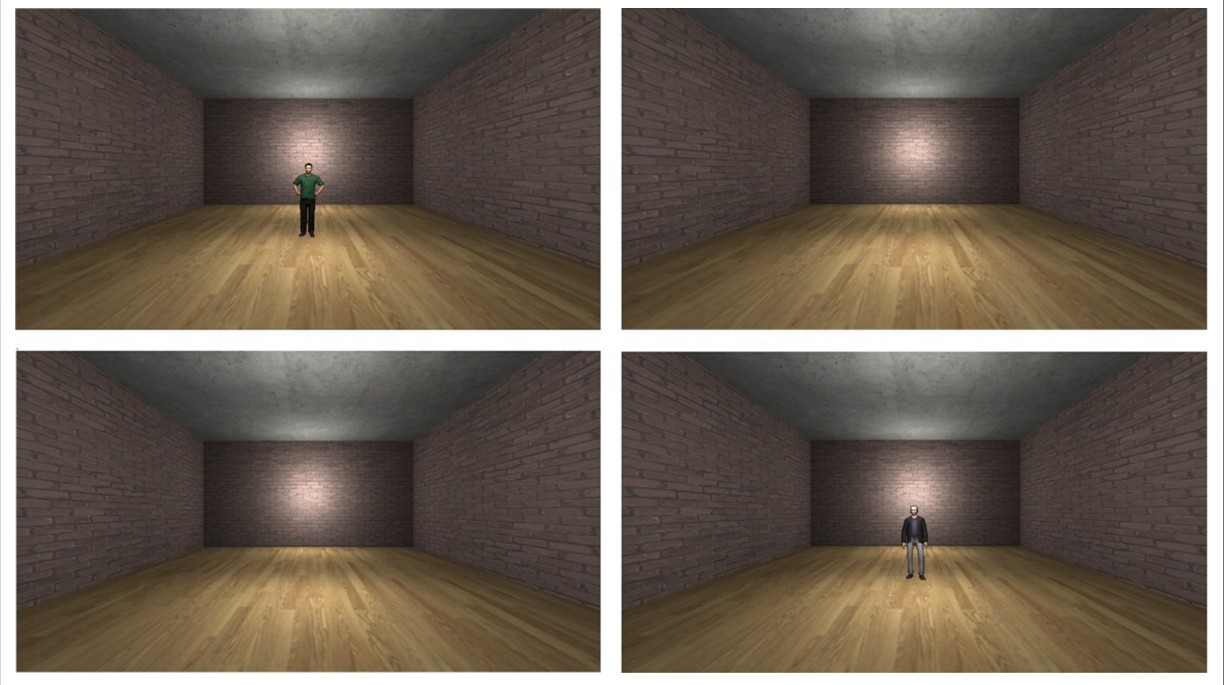

**Figure 3.** Experimental setup. Two male characters were displayed in the scanner during the fear conditioning task (top left, bottom right). One character predicted an electric shock (CS+) whereas the other served as a control stimulus and was never followed by shock (CS-). Between character presentations, participants viewed the empty virtual environment (top right, bottom left).

## Materials

### Stimuli and contexts

During the discriminative fear conditioning task, two male, three-dimensional, virtual humanoid characters and a virtual environment (*Figure 3*) were used. The characters and the environments were created in Unity (version 5.2.3, Unity Technologies, San Francisco, CA) and consisted of a room with four red brick walls, a grey concrete roof, and a wooden floor.

### Stimulus presentation software

The virtual characters and the environment were shown on a flat surface in the MR scanner by a projector (Epson EX5260). Stimulus presentation was handled by the Unity 3D-engine (version 5.2.3, Unity Technologies, San Francisco, CA). The Unity software communicated with BIOPAC (BIOPAC Systems, Goleta, CA) through a custom-made serial interface using standard libraries by Microsoft (Microsoft Corporation, Albuquerque, New Mexico).

### Brain imaging

Imaging data were acquired using a 3.0 T scanner (Discovery MR750, GE Healthcare) and an eight-channel head-coil. Foam wedges, earplugs, and headphones were used to reduce head motion and scanner noise. We acquired T1-weighted structural images with whole-head coverage, TR = 6400ms, TE = 28ms, acquisition time = 6.04 min, and flip angle = 11°. Functional images were acquired using gradient echo-planar-imaging (EPI), TR = 2400ms, TE = 28ms, flip angle = 80°, slice thickness = 3.0 mm with no spacing, axial orientation, frequency direction R/L, interleaved (bottom up), number of slices were 47 and voxel size 3.0 mm$^3$. Higher order shimming was performed, and five dummy scans were run before the experiment.

### Skin conductance responses

Skin conductance was recorded with the MP-150 BIOPAC system (BIOPAC Systems, Goleta, CA). Radio-translucent disposable dry electrodes (EL509, BIOPAC Systems, Goleta, CA) were coated with isotonic gel (GEL101, BIOPAC Systems, Goleta, CA) and placed on the palmar surface of the participant's left hand. The signal was high-pass (0.05 Hz) filtered using the built-in BIOPAC hardware Butterworth filter. SCRs were scored using Ledalab software package (v 3.4.9) (*Benedek and Kaernbach, 2010*) implemented in Matlab 2020a (Mathworks, Inc, Natick, MA). Minimum response threshold was set to 0.01 μS. After filtering and before analysis, the SCR signal was down-sampled from 2000 Hz to 200 Hz (factor mean). The options specified for the Ledalab batch run were 'open', 'biotrace', 'downsample', 10, 'analyze','CDA', 'optimize',4. SCR was analyzed using standard peak score (through-to-peak, TTP.AmpSum) 1–4 s after CS onset for each participant. To check that whole brain SCR correlations were not dependent on the choice of peak scoring window, we also analyzed SCR with a window of 1–5 s after CS onset. We also scored SCR using a software package called PsPM (*Bach and Friston, 2013*), which uses a model-based approach in estimating SCR (see Appendix 8 for details). We performed these variants of SCR scoring as part of a sensitivity analysis to ensure that correlation results between SCR and brain activity were not dependent on the choice of SCR scoring method.

Unlike previous studies considering the neural correlates of CS+ > CS- SCR (*LaBar et al., 1998*; *Phelps et al., 2004*; *Petrovic et al., 2008*; *MacNamara et al., 2015*; *Marin et al., 2020*), SCRs in the present study were range-corrected by Z transformation within individuals (*BenShakhar, 1985*). Z transformation increases sensitivity to conditioning-related effects and prevents confounding by non-conditioning-related individual differences in general SCR magnitude (*BenShakhar, 1985*; *Staib et al., 2015*). For comparison, however, correlations based on raw SCR scores were also examined and can be found in Appendix 3.

Skin conductance was recorded without a low-pass filter. By using this recording procedure, we noticed that a small number of trials produced unreasonably high SCR values (e.g. 17 mS responding), likely due to electrode movement. As such extreme values may skew correlations even using standard scores, we excluded from all analysis trials with a raw value SCR score > 3 mS. This was based on previous research indicating a general maximum SCR between 2 and 3 mS using similar methodology as the one used in this paper (*Boucsein, 2012*). Using this criterion 97/12200 trials were excluded from analysis (0.795% of all trials).

## Procedure

### Fear conditioning task

Two virtual characters served as CSs, one as a threat cue (CS+) predicting the US and the other as a control cue (CS-). CSs were presented on a screen in the MR scanner at a distance of 2.7 m in the virtual environment. The relatively long distance of 2.7 m was selected in order for the effect of conditioning on SCR not to be occluded by proximal threat effects on SCR, as was observed in two previous studies by *Rosén et al., 2017*; *Rosén et al., 2019*. Participants were told prior to the experiment that they could learn to predict the US but were not told which character served as the CS+. Participants were furthermore instructed to indicate which character would be followed by the US by selecting either 'yes' or 'no' by pressing a button immediately following each CS presentation. The inclusion of the button presses was to assure participants' attention during the task as well as to confirm that participants, overall, learned the contingency. Which of the two characters that served as CS+ and CS- was counterbalanced across participants using four different stimulus presentation orders. Each CS presentation lasted for 6 s followed by an inter-stimulus interval of 8–12 s, during which the context was still displayed, but no CS was present.

A habituation phase preceded the fear conditioning, in which each CS was presented four times without reinforcement for a total of eight CS presentations. This was followed by the fear conditioning phase, during which each CS type (CS+ and CS-) was presented 16 times. The experimental task thus consisted of 40 CS presentations in total: 8 during habituation and 32 during conditioning. During conditioning, eight of the CS+ presentations co-terminated with a presentation of the US (50% partial reinforcement schedule; in accordance with guidelines for increasing sensitivity to inter-individual differences in fear conditioning, see *Lonsdorf and Merz, 2017*). Total time for the fear conditioning task was 9 min and 47 s.

The US consisted of an electric shock delivered to the subjects' wrist via radio-translucent dry electrodes (EL509, BIOPAC Systems, Goleta, CA). Prior to the experiment, the shock was calibrated using an ascending staircase procedure wherein shock intensity is increased until rated by participants as 'uncomfortable' but not 'painful' (*Åhs et al., 2015*). US duration was 16ms and controlled using the STM100C module connected to the STM200 constant voltage stimulator (BIOPAC Systems, Goleta, CA).

In all sequences, the first CS+ presentation following the 4 CS+ habituation trials was always reinforced. The sequences differed in whether the CS- or CS+ started the acquisition phase. If the reinforced CS+ is always the first trial in the acquisition phase, the CS- trial following the US will be elevated due to sensitization. This was why the presentation order was counterbalanced.

### Analysis of SCR data

SCR Z scores were averaged separately across CS+ and CS- trials within each participant. A paired samples t test was performed to compare the average CS+ SCRs to the average CS- SCRs at an alpha level (of significance) of $\alpha$ = .05 using JASP software (version 0.14.1, JASP Team (2020)). This allowed us to determine whether the fear conditioning task was successful in evoking greater SCR to the CS+ than to the CS-. Secondly, in order to examine the correlations between conditioned SCR and fMRI responses during fear conditioning, an SCR difference score was calculated for each participant by subtracting the average SCR to CS+ presentations from the average SCR to CS- presentations. The distribution of SCR difference scores was examined to ensure the validity and sensitivity of neural regression analyses (see Appendix 1 regarding methodology and results of SCR distribution analysis).

### Online recording of shock expectancy

Participants pressed one of two buttons each time a CS was displayed to indicate whether they were expecting to receive an electric shock (coded 1) or not (coded 0). The mean response was computed for the CS+ and the CS- presentations. A t-test was performed in JASP to compare mean shock expectancy to the CS+ and CS-, indicating whether participants learned the contingency.

### Analysis of fMRI data

Analyses of fMRI data were performed using SPM12 (version 6685, Wellcome Department of Cognitive Neurology, University College, London). Preprocessing of images included interleaved slice time

correction and realignment of functional volumes. For each participant, the mean functional image was co-registered to the anatomical T1-weigthed image. Quality control of functional images was performed using MRIQC v0.16 (mask validation matching; *Esteban et al., 2017*). Realignment parameters were inspected for excessive movement (defined as 5 mm) during scanning. Anatomical images were segmented using 4 tissue classes and normalized to Montreal Neurological Institute (MNI) standard space. The co-registered functional images were next warped to MNI space using the same parameters that were used for the anatomical image. An 8 mm FHMW Gaussian kernel was used for smoothing of the functional images.

First-level analysis used event-related modeling including regressors for CS+ habituation trials, CS- habituation trials, CS+ acquisition trials, CS- acquisition trials, and US delivery in a general linear model (GLM). Regressors mapped to the intervals of 6 s where each type of stimulus was displayed in the scanner and was convolved with the hemodynamic response function to predict the fMRI time course (for the brief US, the regressor still mapped to the 6 s following US delivery). Also, 6 movement parameter regressors, derived from the image realignment, and a mean value intercept regressor (a vector of ones) were included in the model. At the second level, we first examined the overall whole brain CS+ > CS- BOLD contrast to assess conditioning-related effects. This analysis included both reinforced and non-reinforced CS+ trials, as the US was delivered 2 s after the 4 s sampling window for SCR scoring, and therefore could not confound the SCRs on reinforced trials. Voxel-wise statistical significance was calculated using t tests implemented in the SPM12 software with an alpha level of $\alpha = .05$ using family-wise error (FWE) correction. As results regarding this contrast are not central to the present study, they are published in Appendix 6. However, it should be noted that these results were typical for fear conditioning studies in general (see e.g. *Fullana et al., 2016*). In the present analysis, we proceeded to examine regional activity within the whole brain CS+ > CS- BOLD contrast that covaried with individual differences in the SCR difference score. This was done by entering each participant's previously obtained differential SCR score (CS+ minus CS-) as a second level between-subjects regressor of their average CS+ > CS- BOLD activation, effectively correlating the CS+ > CS- SCR difference with the CS+ > CS- BOLD difference. Voxel-wise statistical significance was again calculated using $t$-tests implemented in the SPM12 software with an FWE corrected alpha level of $\alpha = .05$. Notably, this provided a local maxima height threshold of $t = 4.36$ given our sample size ($N = 285$). This roughly corresponds to a correlation coefficient of $r = .25$ given the conversion formula $r = sqrt (t^2 /(t^2 + DF))$, thus corresponding to an effect size within the medium range according to guidelines by *Cohen, 1988*; *Cohen, 1992*. Such an effect size is lower than estimated effect sizes from the study by *MacNamara et al., 2015*, the only previous study meeting the minimum requirements for sample size suggested by *Yarkoni and Braver, 2010* and performing a correlation analysis similar to the present one. Specifically, by using the conversion formula $r = sqrt (z^2/(z^2 +N))$, we estimated effect sizes in this study to be large ($r = .48$ and $r = .49$). Based on this observation, the power of the present study was considered acceptable.

Secondly, we tested a hypothesized association between individual differences in amygdala response and SCR using a region-of-interest (ROI) analysis. The amygdala ROI was defined using the automated anatomical labeling (AAL version 1) library (*Tzourio-Mazoyer et al., 2002*) and included both right and left amygdala. Analysis was performed the same way as the previous whole brain correlation except that individual SCR scores were this time correlated exclusively to contrast values within the amygdala ROI, thus increasing sensitivity within this theoretically implicated region. This analysis yielded a local maxima height threshold of $t = 3.66$ given our sample size ($N = 285$). This roughly corresponds to a correlation coefficient of $r = 0.21$ given the conversion formula $r = sqrt (t^2 / (t^2 +DF))$, thus corresponding to an effect size within the small to medium range according to guidelines by *Cohen, 1988*; *Cohen, 1992*.

Third, in order to compare the potentially independent contributions to individual differences in conditioned SCR of different clusters of voxels that showed significant correlations in the whole-brain analysis, we extracted the eigenvariates of the first-level CS+ > CS- contrast values in each cluster of voxels. The SPM12 software's built-in 'extract eigenvariates' function was used to extract eigenvariates from unadjusted contrast values. SPM12 uses Singular Value Decomposition (SVD) to calculate eigenvariates. The result was one value per cluster of voxels (see *Table 2* for the list of regions) for each individual. While eigenvariates are strongly correlated ($r > .9$) to the mean contrast value of the same voxels, they are used instead of the mean because they are less sensitive to extreme values in

individual voxels (for more thorough explanations, see *Ridgeway, 2012*; *Penny, 2004*). The vectors of eigenvariates for each region were next used as a regressors of differential SCR scores in a hierarchical regression analysis implemented in the JASP software. The potentially unique contribution to SCR of each region could then be tested by examining the significance of the beta weights of each region within the model.

Finally, in order to determine if the obtained neural correlates to SCR difference scores were driven more by SCR to CS+ or CS-, we correlated eigenvariates extracted from implicated neural regions and average SCRs to the CS+ and CS- separately. In this analysis, we used square root transformed raw value SCRs instead of Z-transformed SCR in order to obtain roughly normalized SCRs without confounding of CS+ and CS- response magnitude, such as that which occurs when using Z transformation. Specifically, since the Z transformed SCR is defined as the difference between the SCR on a given trial (or trial type) and the average SCR across all trials, divided by the standard deviation of SCRs across all trials, this means that using Z scores conflates CS+ and CS- responding with responses to both average CS+ and CS- inclusively, and therefore cannot be used to determine the influence of neural activity on a specific trial type. As the distributions of square root transformed raw value SCRs to the CS+ and CS- still did not meet criteria for normality (by visual inspection of histograms, QQ-plot, and Shapiro-Wilke's test showing $p < 0.001$; see Appendix 2), we used Spearman's *Rho* instead of Pearson's *r* correlations. To compare the difference between CS+ and CS- correlations we used the *Steiger, 1980* direct comparison of dependent correlation coefficients as implemented in free automated software by *Lee and Preacher, 2013*, as this is the most robust way of testing the difference between dependent Spearman coefficients (*Myers and Sirois, 2006*). To compensate for multiple testing during correlation comparisons, we used a Bonferroni corrected alpha level of $\alpha = 0.05/33 = 0.00151$. This compensated for a total of 33 tests, reflecting 11 implicated neural regions each being correlated to CS+ and CS- separately, as well as each having these correlations compared in an additional test.

## Acknowledgements

This research was supported by grants from the Swedish Research Council (2014–01160, 2018–01322) and the Bank of Sweden Tercentenary Foundation (P20-0125).

## Additional information

### Funding

| Funder | Grant reference number | Author |
|---|---|---|
| Swedish Research Council | 2014–01160 | Fredrik Ahs |
| Swedish Research Council | 2018–01322 | Fredrik Ahs |
| Bank of Sweden Tercentenary Foundation | P20-0125 | Fredrik Ahs |

The funders had no role in study design, data collection and interpretation, or the decision to submit the work for publication.

### Author contributions

Kevin Vinberg, Conceptualization, Formal analysis, Visualization, Methodology, Writing - original draft, Writing - review and editing; Jörgen Rosén, Data curation, Software, Formal analysis, Supervision, Investigation, Visualization, Methodology, Writing - review and editing; Granit Kastrati, Project administration, Writing - review and editing; Fredrik Ahs, Conceptualization, Supervision, Funding acquisition, Methodology, Writing - review and editing

### Author ORCIDs

Kevin Vinberg http://orcid.org/0000-0002-4848-3174
Jörgen Rosén http://orcid.org/0000-0002-3688-3859
Granit Kastrati http://orcid.org/0000-0001-9092-4335
Fredrik Ahs http://orcid.org/0000-0002-6355-660X

### Ethics

Participants provided written informed consent in accordance with guidelines from the Regional Ethical Review Board in Uppsala and received SEK 1000 as reimbursement for their participation. The study protocol was approved by the Regional Ethical Review Board (Dnr 2016/171) in Uppsala.

### Decision letter and Author response

Decision letter https://doi.org/10.7554/eLife.69686.sa1
Author response https://doi.org/10.7554/eLife.69686.sa2

## Additional files

### Supplementary files

• Transparent reporting form

### Data availability

The present study reports data from human participants that did not explicitly consent to their raw neuroimaging and physiological data being made public. Therefore, raw neuroimaging and physiological data from the present study cannot currently be made publicly available. Requests for the anonymized raw neuroimaging and physiological data should be made to Fredrik Åhs (fredrik.ahs@miun.se) and will be reviewed by an independent data access committee, taking into account the research proposal and the intended use of the data. Requestors are required to sign a data transfer agreement to ensure participants' confidentiality is maintained prior to release of any data and that procedures conform with the EU legislation on the general data protection regulation and local ethical regulations. While access to raw source data is thus limited, processed data and standardized statistical images sufficient to reproduce the reported results and figures are publicly and freely available at https://osf.io/7dz9p/. Specifically, we provide statistical brain images in NIfTI file format used to render Figure 1a, Figure 2a, Table 2, Appendix 3—figure 1, Appendix 4—figure 1 and Appendix 6—figure 1 of the present study. We also provide brief explanations of the software used to produce all source data files, along with the SPM job files used for neuroimaging analyses. In the event that ethical approval to publicly share the raw neuroimaging data of the present study is obtained at a later stage; this data will also be made publicly available on the OSF site. In the present journal we have included Figure 1—source data 1, which provides source data for Figure 1b, Figure 2b and statistical analyses reported in section 2.2.3 as well as for Appendix 1—figure 1, Appendix 2—figure 1 and Appendix 5—table 1.

The following dataset was generated:

| Author(s) | Year | Dataset title | Dataset URL | Database and Identifier |
|---|---|---|---|---|
| Rosén J, Åhs F, Vinberg K, Kastrati G | 2022 | Genetic influence on proximal, social and conditioned threat responses | https://doi.org/10.17605/OSF.IO/7DZ9P | Open Science Framework, 10.17605/OSF.IO/7DZ9P |

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

## Appendix 1

### Distribution of individual SCR difference scores

Method

The distribution of SCR difference scores was examined using descriptive statistics, visual inspection of histogram, QQ-plot and box plot as well the Shapiro-Wilkes' test (*Shapiro and Wilk, 1965*; *Razali and Wah, 2011*), testing whether the obtained distribution deviated significantly from normality. These analyses were performed to ensure reasonable data with substantial individual differences in conditioned SCR, increasing the reliability and sensitivity of later regression analysis. Notice, however, that normality of the predictor variable is not an assumption in regression analysis using the general linear model (see e.g. *Fox, 2015*).

Results

Shapiro-Wilkes' test ($p < 0.05$; *Shapiro and Wilk, 1965*; *Razali and Wah, 2011*) and a visual inspection of histogram, QQ-plot and box plot revealed a roughly symmetrical distribution (skewness = –0.02, *SE* = 0.14) that deviated from normality due to excessive negative kurtosis (kurtosis = –0.62, *SE* = 0.29; see *Appendix 1—figure 1*). This indicated substantial individual differences in conditioned SCR scores and increased sensitivity of regression analyses.

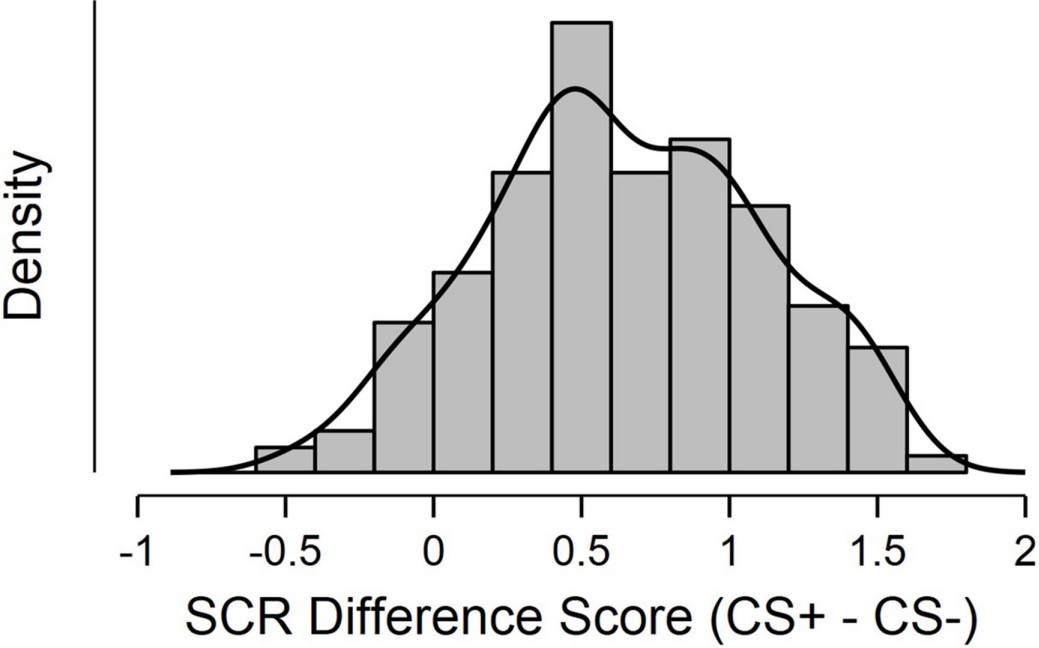

**Appendix 1—figure 1.** Histogram displaying distribution of SCR difference scores, defined as Z transformed average CS+ minus CS- SCR during a fear conditioning paradigm. Sample consisted of 285 participants passing the following exclusion criteria: pregnancy, inability to lie still for a 1 hr duration, intolerance of tight confinements, ongoing psychological treatment, use of psychotropic medications, metal objects in the body (due to surgery, fragmentation etc.), current alcohol or drug related problems. For source data to *Appendix 1—figure 1*, see *Figure 1—source data 1*.

## Appendix 2

### Distribution of square-root transformed raw value SCRs to the CS+ and CS- separately

Method

Similar to the analysis of the distribution of individual differences in SCR difference scores above, we also examined the distribution of square root transformed raw value SCRs to the CS+ and CS-separately. This was done to determine the fit of using Spearman's *rho* correlations or Pearson's *r* correlations in the analyses considering CS+ and CS- responses separately (see Results section 2.2.3). Again, the distribution of SCRs was examined using descriptive statistics, visual inspection of histogram, QQ-plot and box plot as well the Shapiro-Wilkes' test (*Shapiro and Wilk, 1965*; *Razali and Wah, 2011*), testing whether the obtained distribution deviated significantly from normality.

Results

Shapiro-Wilkes' test ($p < 0.001$; *Shapiro and Wilk, 1965*; *Razali and Wah, 2011*) and a visual inspection of histogram, QQ-plot and box plot revealed non-normal distributions due to statistically significant positive skewness (right-tail) for both CS+ (skewness = 0.63, *SE* = 0.14; kurtosis = –0.11, *SE* = 0.29) and CS- (skewness = 0.84, *SE* = 0.14; kurtosis = 0.46, *SE* = 0.29). This indicated the need to use Spearman's *rho* correlations in the correlation analysis of these data.

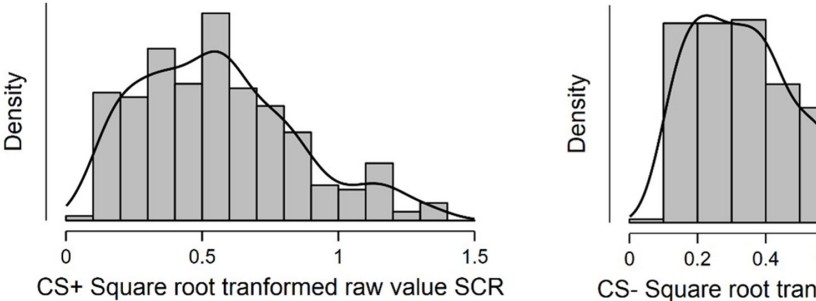

**Appendix 2—figure 1.** Histogram displaying distribution of average square root transformed raw value SCRs to the CS+ (left) and CS- (right) during a fear conditioning paradigm. Sample consisted of 285 participants passing the following exclusion criteria: pregnancy, inability to lie still for a 1 hr duration, intolerance of tight confinements, ongoing psychological treatment, use of psychotropic medications, metal objects in the body (due to surgery, fragmentation etc.), current alcohol or drug related problems. For source data to *Appendix 2—figure 1*, see *Figure 1—source data 1*.

## Appendix 3

### Whole brain correlation analysis using square root transformed raw value SCR

In order to examine if our choice of using range correction by Z transformation affected our results, we repeated our whole brain analysis using square-root transformed raw value SCR such as in previous analyses (e.g. *MacNamara et al., 2015*; *Marin et al., 2020*). Results demonstrated an almost identical pattern of activation to that in the main analysis of the main text, implicating dorsal anterior cingulate cortex/anterior midcingulate cortex, right anterior insula, right inferior frontal gyrus/frontal operculum, bilateral temporoparietal junction/superior temporal gyrus, right superior frontal gyrus/dorsal premotor cortex and a right-lateralized midbrain region in areas consistent with periaqueductal gray and reticular formation (see *Appendix 3—figure 1* and *Appendix 3—table 1*). In comparison to the main results presented in *Table 2* of the main text, this analysis thus implicated the same set of regions except not the left superior frontal gyrus/dorsal premotor cortex and right superior parietal lobe. Notably, this was two of the three smallest clusters of activation in the main analysis. We also notice that the right superior frontal gyrus cluster was larger than when using Z transformation (125 voxels instead of 44 voxels) and this time also overlapped with areas consistent with the right supplementary motor area. In summary, however, we conclude that our choice of using Z transformation had limited effect on our results.

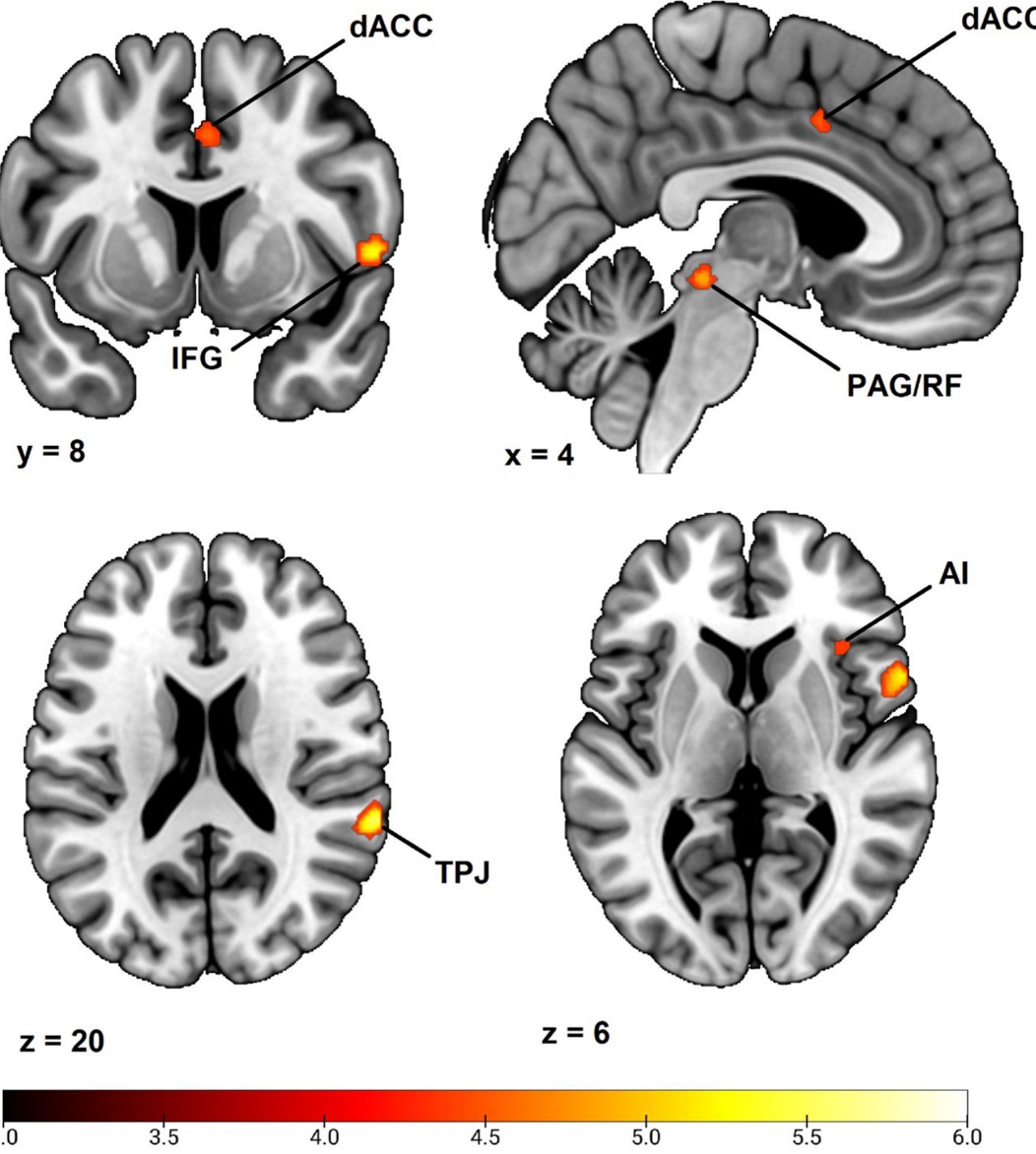

**Appendix 3—figure 1.** Correlation between individual differences in conditioned SCR and whole brain responses during fear conditioning, obtained using individual square root transformed raw value SCR scores (average CS+ minus average CS- SCR) as a second level between-subjects regressor of the average CS+ > CS- BOLD activation in SPM12 (Wellcome Centre for Human Neuroimaging, University College, London) software. Sample consisted of 285 participants passing the following exclusion criteria: pregnancy, inability to lie still for a 1 hr duration, intolerance of tight confinements, ongoing psychological treatment, metal objects in the body (due to surgery, fragmentation etc.), ongoing substance abuse, use of psychotropic medications, unsuccessful recording of skin conductance responses, loss of brain imaging data due to excessive head movement, participant failure to comply with task instruction regarding button press in at least 80% of trials. Displayed is an activation map of key implicated neural regions. Color-coded t values ranges from t = 3 to t = 6. The statistical image was thresholded at p < 0.05 FWE-corrected and displayed on an anatomical brain template. dACC = dorsal Anterior Cingulate Cortex. TPJ = Temporoparietal Junction. IFG = Inferior Frontal Gyrus. PAG/RF = Periaqueductal gray/Reticular Formation. AI = Anterior Insula.

**Appendix 3—table 1.** Whole Brain Correlation to Conditioned SCR using square root transformed raw value SCR.

| Anatomical region | Hemisphere | Voxels | t | MNI Coordinates | | |
|---|---|---|---|---|---|---|
| | | | | x | y | z |
| Dorsal Anterior Cingulate Cortex/Anterior Midcingulate Cortex | N/A | 21 | 4.58 | 4 | 8 | 40 |
| Anterior Insula | Right | 18 | 4.60 | 36 | 22 | 6 |
| Inferior Frontal Gyrus/Frontal Operculum | Right | 144 | 5.66 | 56 | 12 | 2 |
| Temporoparietal Junction/Superior Temporal Gyrus | Right | 116 | 5.64 | 64 | –40 | 20 |
| Superior Frontal Gyrus/Dorsal Premotor Cortex/Supplementary Motor Area | Right | 125 | 5.54 | 16 | 0 | 66 |
| Midbrain | Right | 61 | 5.03 | 6 | –30 | –10 |
| Temporoparietal Junction/Superior Temporal Gyrus | Left | 6 | 4.49 | –62 | –36 | 22 |

Note. MNI coordinates and t values represent significant peak voxels of each cluster. Statistical significance was calculated using t tests implemented within the SPM software with an FWE corrected alpha level of $\alpha = .05$.

Directly comparing SCR scores obtained using either Z transformation or square root transformed raw value SCRs demonstrated a high correlation between them (Spearman's rho $r_s = 0.86$; $p < 0.001$). Furthermore, Z transformed conditioned SCR scores were shown to correlate with raw response magnitude to both CS+ ($r_s = 0.62$; $p < 0.001$) and CS- ($r_s = 0.19$; $p = 0.001$), meaning that individuals who showed greater differentiation between CS+ and CS- also had overall higher magnitude SCR.

## Appendix 4

## Whole brain correlation analysis with minimal participant exclusion (*N* = 303)

In order to examine if our choice of participant exclusion in the main analysis affected our results, we repeated our whole brain analysis using the full sample of participants with both fMRI and SCR data. In addition to the 285 participants in the main analysis, this included an additional 5 participants with excessive head movement, an additional 11 participants that failed to comply with the task instruction regarding button presses in at least 80% of trials and an additional 7 participants that used psychotropic medication. Thus, this full sample size consisted of 303 participants.

Results using *N* = 303 demonstrated an almost identical pattern of activation to that in the main analysis of the main text, implicating the dorsal anterior cingulate cortex/anterior midcingulate cortex, right anterior insula, right inferior frontal gyrus/frontal operculum, right temporoparietal junction/superior temporal gyrus, right superior frontal gyrus/dorsal premotor cortex, right midbrain in areas consistent with periaqueductal gray and reticular formation and a small regional activation in left inferior frontal gyrus (see *Appendix 4—figure 1* and *Appendix 4—table 1*). In comparison to the main results presented in *Table 2* of the main text, this analysis thus implicated the same set of regions except the left superior frontal gyrus/dorsal premotor cortex, left temporoparietal junction and right superior parietal lobe. Notably, this was the three smallest clusters of activation in the main analysis. While this analysis using *N* = 303 also implicated a new neural region, namely the left inferior frontal gyrus, this finding does not appear as robust as previous findings and should therefore be interpreted with caution.

**Appendix 4—table 1.** Whole Brain Correlation to Conditioned SCR without participant exclusion (*N* = 303).

| Anatomical region | Hemisphere | Voxels | t | MNI Coordinates | | |
|---|---|---|---|---|---|---|
| | | | | x | y | z |
| Dorsal Anterior Cingulate Cortex/Anterior Midcingulate Cortex | N/A | 101 | 5.06 | 6 | 10 | 38 |
| Anterior Insula | Right | 16 | 4.54 | 36 | 20 | 6 |
| Inferior Frontal Gyrus/Frontal Operculum | Right | 149 | 5.67 | 56 | 10 | 2 |
| Temporoparietal Junction/Superior Temporal Gyrus | Right | 91 | 5.82 | 64 | –40 | 20 |
| Superior Frontal Gyrus/Dorsal Premotor Cortex/Supplementary Motor Area | Right | 17 | 4.72 | 18 | 0 | 68 |
| Midbrain | Right | 74 | 5.29 | 10 | –32 | –10 |
| Inferior frontal gyrus | Left | 2 | 4.35 | –56 | 0 | 2 |

Note. MNI coordinates and t values represent significant peak voxels of each cluster. Statistical significance was calculated using t tests implemented within the SPM software with an FWE corrected alpha level of $\alpha$ = .05.

In summary, both supplementary whole brain analyses (Appendix 3 and 4) implicated largely the same set of regions as the main analysis of the main text (see *Table 2*). In particular, right-lateralized regional activations in the dorsal anterior cingulate cortex/anterior midcingulate cortex, anterior insula, inferior frontal gyrus, temporoparietal junction, superior frontal gyrus/dorsal premotor cortex and midbrain were consistent across analyses. Thus, findings regarding these regions appear robust. However, neural activations in the left superior frontal gyrus/dorsal premotor cortex, left temporoparietal junction, left inferior frontal gyrus and right superior parietal lobe were not consistently implicated. Hence, findings regarding these latter regions do not appear as robust. However, two important things should be noticed regarding these latter regions. First, our conclusions regarding the main findings of the study are not heavily dependent on the activation of these latter regions. Second, for reasons explained in the Materials and Methods section of the main text, we consider the main analysis of the main text to be the most valid analysis.

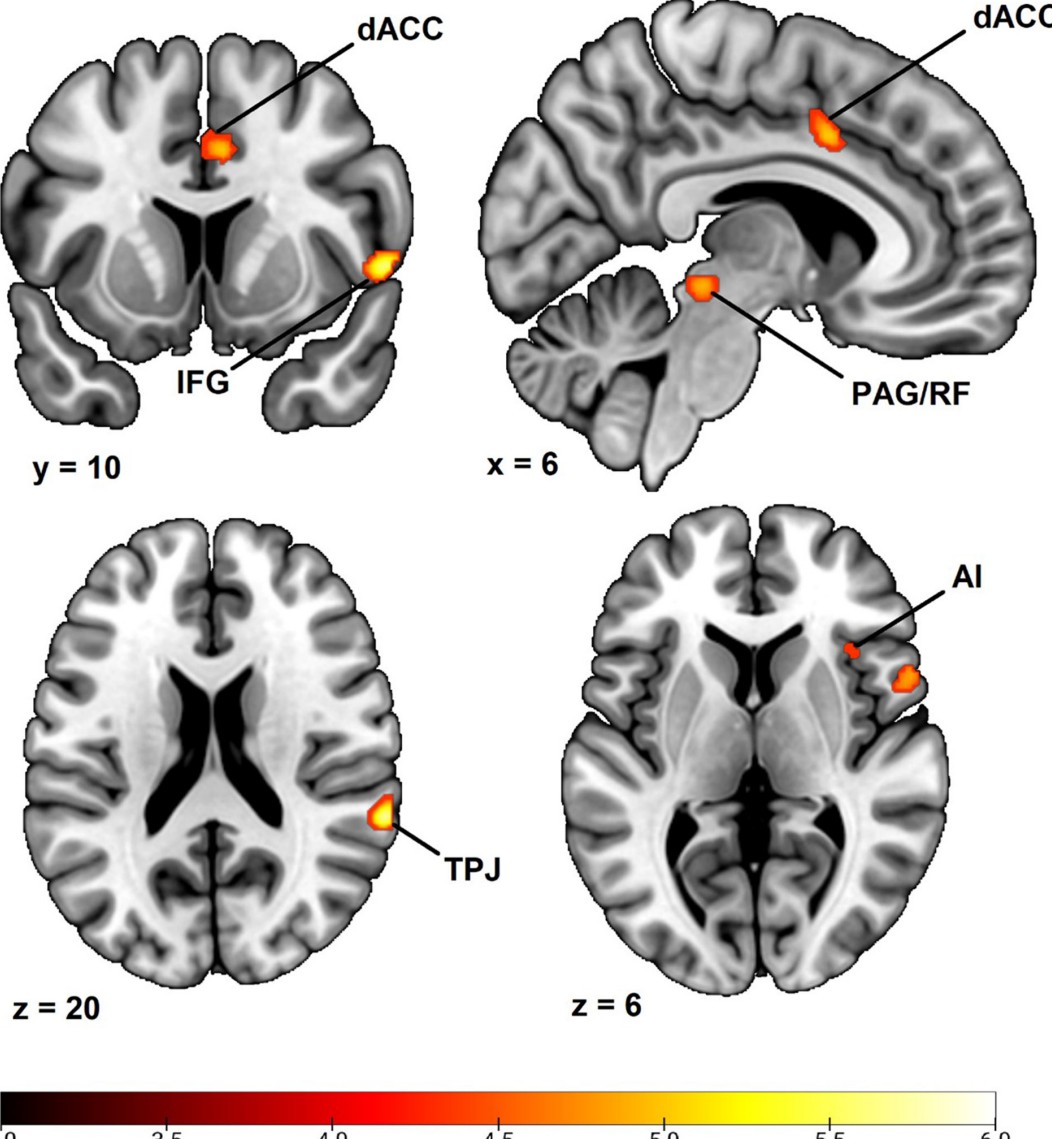

**Appendix 4—figure 1.** Correlation between individual differences in conditioned SCR and whole brain responses during fear conditioning with minimal participant exclusion. Results obtained using individual SCR scores (Z transformed average CS+ minus CS- SCR) as a second level between-subjects regressor of the average CS+ > CS- BOLD activation in SPM12 (Wellcome Centre for Human Neuroimaging, University College, London) software. Sample consisted of 303 participants passing the following exclusion criteria: pregnancy, inability to lie still for a 1 hr duration, intolerance of tight confinements, ongoing psychological treatment, metal objects in the body (due to surgery, fragmentation etc.), ongoing substance abuse. Note that this is a larger sample than in the main analyses of the main text (*n* = 285), including an additional 5 participants with loss of brain imaging data due to excessive head movement, an additional 11 participants that failed to comply with the task instruction regarding button presses in at least 80% of trials and an additional 7 participants that used psychotropic medication. Displayed is an ctivation map of key implicated neural regions. Color-coded *t* values ranges from *t* = 3 to *t* = 6. The statistical image was thresholded at *p* < 0.05 FWE-corrected and displayed on an anatomical brain template. *R* = Right. dACC = dorsal Anterior Cingulate Cortex. TPJ = Temporoparietal Junction. IFG = Inferior Frontal Gyrus. PAG/SC = Periaqueductal gray/Superior Colliculus. AI = Anterior Insula.

# Appendix 5

## Beta coefficients from regression model predicting individual differences in conditioned SCR

**Appendix 5—table 1.** Beta coefficients from regression model.

| Model | | Unstandardized | Standard Error | Standardized | t | p |
|---|---|---|---|---|---|---|
| $H_0$ | (Intercept) | 0.642 | 0.028 | | 23.284 | <.001 |
| $H_1$ | (Intercept) | 0.434 | 0.044 | | 9.773 | <.001 |
| | R IFG | 0.088 | 0.086 | 0.105 | 1.026 | 0.306 |
| | R TPJ | 0.136 | 0.088 | 0.133 | 1.551 | 0.122 |
| | R Midbrain | 0.178 | 0.107 | 0.122 | 1.671 | 0.096 |
| | R dPMC | 0.077 | 0.105 | 0.065 | 0.733 | 0.464 |
| | dACC | –0.064 | 0.092 | –0.072 | –0.699 | 0.485 |
| | R AI | 0.000 | 0.080 | 0.000 | 0.011 | 0.992 |
| | R SPL | 0.039 | 0.062 | 0.047 | 0.618 | 0.537 |
| | L SFG | 0.079 | 0.085 | 0.075 | 0.939 | 0.349 |
| | L TPJ | –0.011 | 0.070 | –0.013 | –0.162 | 0.872 |

Note. Coefficient results obtained from regression analysis within the JASP software (JASP Team (2020). JASP (Version 0.14.1) [Computer software]) using eigenvariates from implicated whole brain regions as independent regressors of individual differences in conditioned SCR. Abbreviations: R = Right; L = Left; TPJ = Temporoparietal Junction; IFG = Inferior Frontal Gyrus; dACC = dorsal Anterior Cingulate Cortex; dPMC = dorsal Premotor Cortex; AI = Anterior Insula; SPL = Superior Parietal Lobe.

## Appendix 6

### Whole brain CS+ > CS- BOLD contrast

Examining the whole brain CS+ > CS- BOLD contrast revealed a pattern of activation typical to fear conditioning studies in general (*Appendix 6—figure 1*; *Appendix 6—table 1*; for comparison see *Fullana et al., 2016*). A large major cluster (46343 voxels) was centered on bilateral insula spreading laterodorsally to bilateral inferior, middle, superior and precentral frontal gyri, medially to ventral striatum, caudally to thalamus and adjacent midbrain/brainstem regions (including the periaqueductal gray, reticular formation and dorsal pons), as well as spreading dorsally to medial wall cortex (including dACC, midcingulate cortex, pre-supplementary and supplementary motor areas and dorsal anterior precuneus). The cluster also encompassed large activation of bilateral middle and superior temporal gyri and supramarginal gyrus as well as inferior and superior parietal cortex. Also included in the cluster was bilateral amygdala activation as well as small portions of cerebellum. In separate clusters we observed activation in bilateral cuneus, bilateral middle prefrontal cortex and bilateral visual cortex.

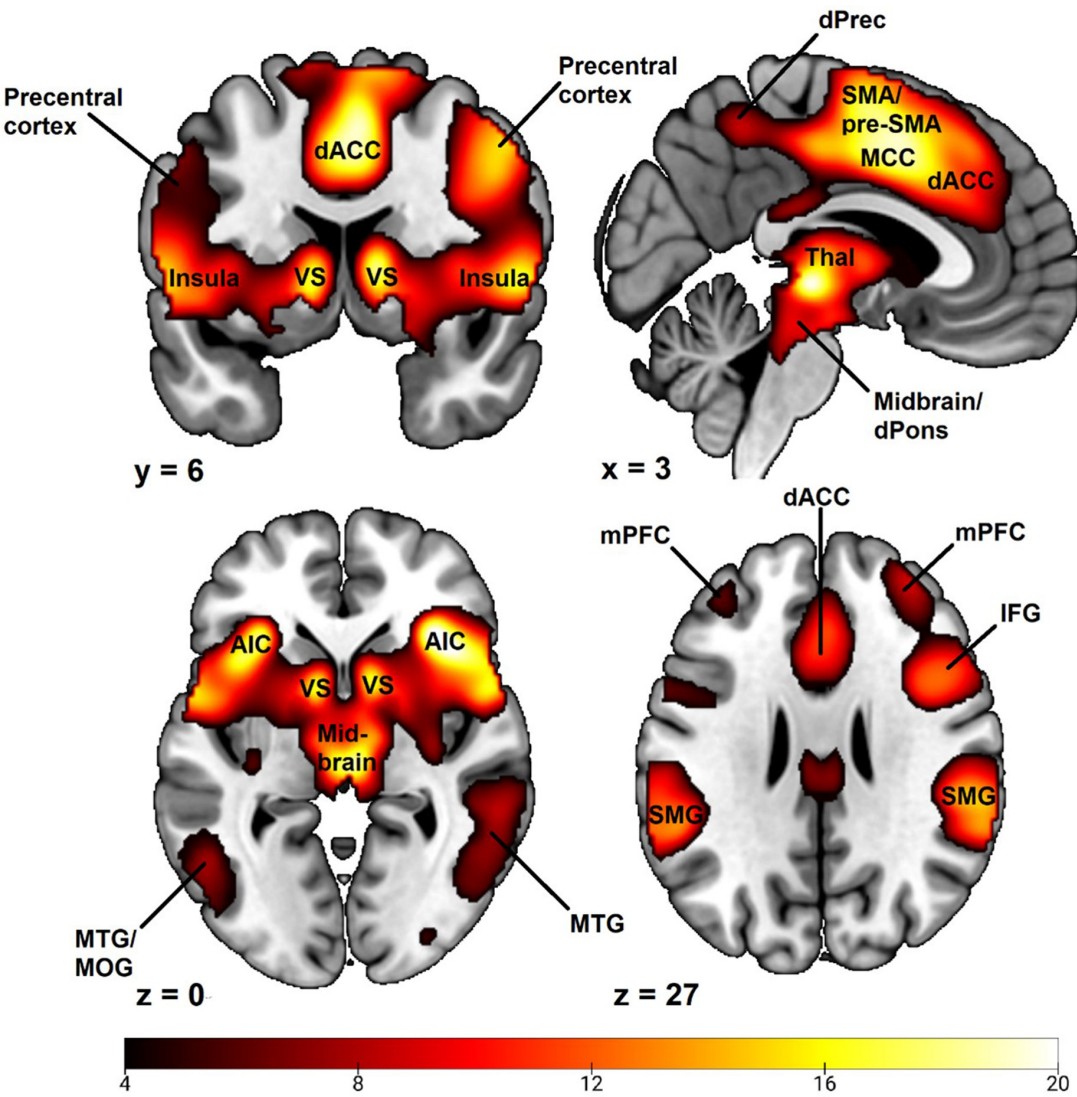

**Appendix 6—figure 1.** Whole brain CS+ > CS- BOLD contrast activations during a fear conditioning paradigm. Results obtained using first-level event-related modeling of conditioned and unconditioned stimuli in a general linear model predicting BOLD signal and then examining the overall whole brain CS+ > CS- BOLD contrast on the group level in SPM12 (Wellcome Centre for Human Neuroimaging, University College, London). Sample consisted of 285 participants passing the following exclusion criteria: pregnancy, inability to lie still for a 1 hr duration,

*Appendix 6—figure 1 continued*
intolerance of tight confinements, ongoing psychological treatment, metal objects in the body (due to surgery, fragmentation etc.), ongoing substance abuse, use of psychotropic medications, unsuccessful recording of skin conductance responses, loss of brain imaging data due to excessive head movement, participant failure to comply with task instruction regarding button press in at least 80% of trials. Color-coded *t* values ranges from *t* = 4.0 to *t* = 20. Statistical images are thresholded at *p* < 0.05 FWE-corrected. Abbreviations: dACC = dorsal anterior cingulate cortex; VS = ventral striatum; SMA = supplementary motor area; Pre-SMA = pre-supplementary motor area; dPrec = dorsal precuneus; MCC = midcingulate cortex; Thal = thalamus; dPons = dorsal pons; AIC = anterior insula cortex; MTG = medial temporal gyrus; MOG = medial occipital gyrus; mPFC = middle prefrontal cortex; MFG = middle frontal gyrus; IFG = inferior frontal gyrus; SMG = supramarginal gyrus.

**Appendix 6—table 1.** Whole brain CS+ > CS- BOLD contrast activations.

| Cluster | Brain regions with peaks within cluster | t | Peak MNI Coordinates | | |
| --- | --- | --- | --- | --- | --- |
| | | | x | y | z |
| **Cluster 1 (46343 voxels):** insula, ventral striatum, inferior, middle and superior frontal cortex, thalamus, midbrain/brainstem, anterior and midcingulate cortex, supplementary motor area, middle and superior temporal gyri, supramarginal gyrus, inferior and superior parietal cortex, amygdala, cerebellum. | - | > 4.34 | - | - | - |
| | Right Insula | 21.71 | 36 | 24 | 2 |
| | Left Insula | 18.71 | −32 | 24 | -4 |
| | Left Insula | 17.55 | −40 | 18 | -4 |
| | Right Frontal Operculum | 17.06 | 56 | 6 | 2 |
| | Supplementary Motor Area | 19.24 | 4 | 6 | 50 |
| | Supplementary Motor Area/Midcingulate Cortex | 19.21 | 2 | 8 | 46 |
| | Right Caudate | 18.35 | 10 | 6 | 4 |
| | Left Caudate | 17.20 | −10 | 6 | 2 |
| | Thalamus | 20.10 | 4 | −24 | -2 |
| | Right Temporoparietal Junction | 17.86 | 50 | −32 | 20 |
| | Right Precentral Gyrus | 16.29 | 46 | 2 | 46 |
| **Cluster 2 (217 voxels):** Right Cuneus | | | | | |
| | Right Cuneus | 8.46 | 14 | −72 | 38 |
| **Cluster 3 (208 voxels):** Left Middle Prefrontal Cortex | | | | | |
| | Left Middle Prefrontal cortex | 6.33 | −38 | 46 | 24 |
| **Cluster 4 (69 voxels):** Left Cuneus, Left Posterior Precuneus | Left Posterior Precuneus | 6.04 | −10 | −74 | 38 |
| **Cluster 5 (61 voxels):** Left dorsal Cerebellum, Left Fusiform Gyrus | Left Cerebellum VI | 5.61 | −34 | −60 | −26 |
| | Left Cerebellum VI | 5.05 | −38 | −64 | −24 |
| | Left Fusiform Gyrus | 5.00 | −40 | −66 | −20 |
| | Left Cerebellum VI | 4.45 | −38 | −54 | −26 |
| **Cluster 6 (15 voxels):** Right Visual Cortex | Right Visual Cortex | 4.96 | 20 | −64 | 6 |
| **Cluster 7 (9 voxels):** Left Visual Cortex | Left Visual Cortex | 4.69 | −14 | −70 | 4 |
| **Cluster 8 (8 voxels):** Right Frontal Pole/Superior Orbital Gyrus | Right Frontal Pole | 4.56 | 26 | 60 | -6 |

*Note.* MNI coordinates and *t* values represent significant peak voxels within each cluster. Statistical significance was calculated using t tests with an FWE corrected alpha level of $\alpha$ = .05 within the SPM software.

# Appendix 7

## Sensitivity analysis

**Appendix 7—table 1.** Sensitivity analysis.

Correlations to conditioned SCR in the significant peak voxels from the main analysis altering SCR definitions, covariates, and sample selection. Pearson correlation coefficients (r) were included to permit comparison of effect sizes. *Main analysis* refers to the analysis presented in the results section of the paper. *Square-root transformed* refers to using square-root transformed Z-transformed SCRs. The *PsPM analysis* used a model-based approach to compute SCR based on a Dynamic Causal Modeling framework. Please notice that this model may not be optimally suited to our data, see Appendix 8 for more information. *5 s SCR window* used 5 s time window following CS-onset for peak-detection. *Non-reinforced trials* only included non-reinforced CS+ trials when computing SCR and neural responses. To *control for shock expectancy effects*, we included average shock expectancy (rated online as 0 or 1) as a covariate correlating with whole-brain responses. We also controlled for eventual familial influences by splitting twin pairs and correlating SCR in the two samples (first twin, second twin) to brain contrast values. Note that the full sample included twin pairs as well as twins without a sibling, hence there is a discrepancy in the number of twins for first and second columns (135 vs 146). Correlations between SCR to US and CS+ > CS- contrast-values are shown in the last column for reference.

| Anatomical region | Hemisphere | Voxels | x | y | z | Main analysis $t_{283}$ (r) | | Square-root transformed $t_{283}$ (r) | | PsPM (DCM) analysis $t_{283}$ (r) | | 5 s SCR time-window $t_{283}$ (r) | | Non-reinforced trials $t_{283}$ (r) | | Control for shock expectancy $t_{282}$ (r) | | Excluding first twin $t_{135}$ (r) | | Excluding second twin $t_{146}$ (r) | | SCR to US $t_{283}$ (r) | |
|---|---|---|---|---|---|---|---|---|---|---|---|---|---|---|---|---|---|---|---|---|---|---|---|
| Dorsal Anterior Cingulate Cortex/Anterior Midcingulate Cortex | N/A | 50 | 6 | 8 | 40 | 4.79 | (0.27) | 4.58 | (0.26) | 3.63 | (0.21) | 4.95 | (0.28) | 4.55 | (0.26) | 3.07 | (0.18) | 3.90 | (0.32) | 2.78 | (0.23) | 0.00 | (0.00) |
| Anterior Insula | Right | 20 | 36 | 20 | 6 | 4.65 | (0.27) | 4.50 | (0.26) | 4.51 | (0.26) | 5.01 | (0.29) | 3.95 | (0.23) | 2.69 | (0.16) | 3.68 | (0.30) | 2.88 | (0.24) | 0.31 | (0.02) |
| Inferior Frontal Gyrus/Frontal Operculum | Right | 138 | 56 | 10 | 2 | 5.80 | (0.33) | 5.31 | (0.30) | 5.03 | (0.29) | 5.94 | (0.33) | 5.02 | (0.29) | 3.44 | (0.20) | 4.69 | (0.37) | 3.59 | (0.30) | 1.35 | (0.08) |
| Temporoparietal Junction/Superior Temporal Gyrus | Right | 81 | 64 | -40 | 20 | 5.66 | (0.32) | 5.45 | (0.31) | 4.39 | (0.25) | 5.68 | (0.32) | 4.49 | (0.26) | 3.54 | (0.21) | 4.22 | (0.34) | 3.81 | (0.31) | 1.44 | (0.09) |
| Superior Frontal Gyrus/Dorsal Premotor Cortex | Right | 44 | 18 | 0 | 68 | 5.21 | (0.30) | 5.25 | (0.30) | 4.12 | (0.24) | 5.20 | (0.30) | 4.73 | (0.27) | 2.58 | (0.15) | 3.70 | (0.30) | 3.65 | (0.30) | 0.91 | (0.05) |
| Midbrain | Right | 59 | 10 | -30 | -12 | 5.22 | (0.30) | 5.14 | (0.29) | 5.06 | (0.29) | 5.03 | (0.29) | 4.53 | (0.26) | 2.78 | (0.16) | 4.00 | (0.33) | 3.31 | (0.27) | 2.51 | (0.15) |
| Superior Parietal Lobe/Postcentral Gyrus | Right | 3 | 22 | -46 | 70 | 4.52 | (0.26) | 4.28 | (0.25) | 3.89 | (0.23) | 4.97 | (0.28) | 4.30 | (0.25) | 2.89 | (0.17) | 3.51 | (0.29) | 2.86 | (0.24) | 1.81 | (0.11) |
| Superior Frontal Gyrus/Dorsal Premotor Cortex | Left | 2 | -14 | -2 | 72 | 4.46 | (0.26) | 4.39 | (0.25) | 4.12 | (0.24) | 4.71 | (0.27) | 4.45 | (0.26) | 3.05 | (0.18) | 4.52 | (0.36) | 2.00 | (0.17) | 0.56 | (0.03) |

*Appendix 7—table 1 continued on next page*

*Appendix 7—table 1 continued*

| Anatomical region | Hemisphere | Voxels | x | y | z | Main analysis t₂₈₃ (r) | | Square-root transformed t₂₈₃ (r) | | PsPM (DCM) analysis t₂₈₃ (r) | | 5 s SCR time-window t₂₈₃ (r) | | Non-reinforced trials t₂₈₃ (r) | | Control for shock expectancy t₂₈₂ (r) | | Excluding first twin t₁₃₅ (r) | | Excluding second twin t₁₄₆ (r) | | SCR to US t₂₈₃ (r) | |
|---|---|---|---|---|---|---|---|---|---|---|---|---|---|---|---|---|---|---|---|---|---|---|---|
| Temporoparietal Junction/Superior Temporal Gyrus | Left | 1 | −62 | −36 | 22 | 4.37 | (0.25) | 4.24 | (0.24) | 3.02 | (0.18) | 5.07 | (0.29) | 3.93 | (0.23) | 1.81 | (0.11) | 3.73 | (0.31) | 2.55 | (0.21) | 1.65 | (0.10) |

Note. MNI coordinates and t values represent significant peak voxels of clusters from the main analysis. Statistical significance was calculated using t tests implemented within the SPM software with an FWE corrected alpha level of α = .05.

## Appendix 8

### Complementary analyses of SCR using a model-based approach in the PsPM software

Method The Ledalab software package v 3.4.9, *Benedek and Kaernbach, 2010* used for our main SCR analysis has been shown to sometimes yield unpredictable results, especially compared to standard peak scoring methods (*Bach, 2014*). Thus, we decided to replicate our results using a different software package (PsPM v 5.1.1, *Bach et al., 2010*). Before analysis, data was imported into a data structure created by the import function in PsPM. A 6 s delay between CS and US onset was entered and recorded trigger times were specified for each event. We then modeled the data based on a non-linear dynamic causal model (DCM). Data was normalized, but the rest of the options were set to default. This corresponds to a "full interval" model with flexible onset of conditioned responding during the full CS duration (see *Kuhn et al., 2022*, for a discussion). In the first-level, four conditions were specified: CS+ > CS-, CS+, CS- and US. Results were then extracted using the export statistics function in PsPM.

Notably, this model may not be optimally suited for fear conditioning data with a 6 s CS duration. This is because modeling the conditioned response using a flexible onset during the full CS duration may conflate CS responding with US responding, in particular at longer CS durations (cf. page 23, manual for PsPM 6.0.0, available at http://pspm.sourceforge.net/; see also *Kuhn et al., 2022*). Therefore, this analysis should only be viewed as an additional sensitivity analysis complimentary to our main analysis, showing converging results across different types of SCR scoring. For researchers looking to use this model in their future research, we recommend consulting the PsPM manual as well as the empirical investigation by *Kuhn et al., 2022*.

