## [Editor Report]

Vinberg et al. provide a conceptual replication on individual differences in conditioned skin conductance response during fear acquisition training and BOLD fMRI in a large sample (*N* = 285) of healthy individuals (mono- and dizygotic twins). The authors report results that are in line with previous work and new results from a whole-brain analysis and suggest unique and shared contributions of individual brain regions.

---

## [Decision Letter]

**Decision letter after peer review:**

Thank you for submitting your article "Whole Brain Correlates of Individual Differences in Skin Conductance Responses during Human Fear Conditioning" for consideration by *eLife*. Your article has been reviewed by 3 peer reviewers, and the evaluation has been overseen by Drs. Shackman (Reviewing Editor) and Büchel (Senior Editor).

Essential Revisions

• Authors need to better integrate genetic relatedness into their report.

o The study sample is relatively large (*N* = 285) – yet the sample is special in that participants were genetically related and as siblings (twins, mono- and dizygotic twins) shared environmental influences. This become only apparent in the method section (and discussion?) but needs to be mentioned upfront in the abstract, intro and included in the discussion as this may have an impact on the results. From the methods section it remained unclear how many pairs of di- and monozyotic twins were included in the study and more information on the sample (age range for instance) would be desirable.

• Polish the manuscript. There are grammatical errors. Wording and clarity could be improved. If the manuscript is meant for neuroscientists and psychologists in general (not only human fear conditioning experts), the reader probably needs some more background on some of the topics dealt with.

• The authors need to be more precise and nuanced in their description of prior work (see below)

• Introduction

o Provide a less superficial review of the current state of the science. Replication attempts are most useful when it is clearly outlined which effect is aimed to be replicated, a thorough and precise status quo of the literature is provided and in case of conceptual replications which procedural and analytical specifications differ from the previous, to-be-replicated work. It would be helpful for the reader if the exact results of previous work are, the employed procedures and analyses of previous work were described and discussed in relation to the present work in more detail.

o Authors need to better clarify the innovation/novelty of the aims and approach

o Provide a stronger motivation for the amygdala focus

– One of the Reviewers noted that, as it is currently written, I found the emphasis on the amygdala problematic. One of the goals of the manuscript is "to replicate previous findings of an association between individual differences in amygdala response and SCR using an ROI approach". Why is this a goal? Isn't the goal to understand the brain correlates of differences in human conditioning. Maybe the ROI result can be added as an additional result, but it probably should not be part of the goals, at least without much stronger justification in the Introduction.

– Another Reviewer noted that, Finding out about an SCR-amygdala BOLD correlation is one of the motives of this study. I was left unsure why mass-univariate amygala activity should correlate with the CS+/CS- difference. According to Fullana et al. (2016), there is no evidence of group-level amygdala activity in the CS+/CS- contrast. On the other hand, patterns of neural responses in the amygdala distinguish CS+/CS- (Bach et al. 2010 J Neurosci, Visser et al. 2011 J Neurosci, Staib and Bach 2018 NIMG). CS+-on neurons are sparse in the amygdala (Reijmers et al. 2007 Science) and is an equal number of CS+ on and CS+ off neurons in the central amygdala (Tovote et al. 2015 Nat Rev Neurosci and original papers referenced therein). On the balance of things, the motivation for looking at amygdala activity in the first place is weak. This needs to be better motivated in light of the available evidence

• Approach

o Authors need to clarify the approach and provide some crucial missing details that have the potential to markedly influence the results and conclusions

– Enrollment criteria

• Some inclusion/exclusion criteria are not well defined (e.g., "current alcohol or drug-related problems") or unclear (e.g., why should someone receiving psychological treatment be excluded? were only psychotropic medications -and not other medications- excluded? )

– SCR/EDA

• The peak-scoring windows for the SCR analysis are unclear, and potentially quite problematic. This, together with the comparably large effect size for the CS+/CS- difference in SCR, suggests a potential risk that the authors may have inadvertently looked at outcome-driven (US- or omission-driven) SCR, rather than conditioned SCR. This would call into question the brain-behavior associations

o The authors seem to use a 6 s-SOA delay fear conditioning paradigm. SCR scoring was done with ledalab, using the "maximum phasic driver amplitude 1-4 seconds after CS presentation for each participant". The potential problem is the peak detection window. First, can the authors clarify whether the peak window is 1-4 s after CS onset or after CS offset? Second, do they analyse only non-reinforced trials or also reinforced trials?

o What Ledalab calls the "driver" is a peripheral neural impulse at some unspecified place in the peripheral autonomic system. As can be seen in figure 5a in Benedek and Kaernbach (2010) where SCR were elicited by external events, this "driver" peaks around 2 s after an external event. So, if the US (or US omission) elicits an SCR, then the estimated "driver" will peak 2 s after CS offset and would be included in a 1-4 s window after CS offset. If, on the other hand, there was a gradual increase in SCR > 2 s into the CS, then the driver would peak > 4 s after CS onset and would not be included in a 1-4 s window after CS onset.

o In sum, the authors need to better work out and explain their peak scoring windows. They should also compare reinforced and non-reinforced CS+ trials, to rule out any bias in their analysis. Given that Ledalab yields no better results than standard peak scoring, and sometimes worse results (Bach 2014 Biological Psychology), they may want to consider using a standard peak-scoring analysis or similar strategy. (I note that the standard procedures implemented in PsPM – Bach et al. 2020 Beh Res Therapy – are not optimized for this 6-s SOA, even though there is an option that makes the models suitable for this case as well.)

• In general, the authors need to provide more details on the SCR data acquisition, processing, and analyses (e.g. the versions of the software used and specific settings/parameters e.g., sampling rate, all filters, downsampling if any, derivative scores, quality assurance/control procedures, etc.).

• The pre-processing (filtering) is not sufficiently justified. Authors may want to consider the results by Privratsky et al. 2020 (https://pubmed.ncbi.nlm.nih.gov/33075428/) to guide their choice of filters

• The authors need to elaborate on the advantages of using Z-transformed SCR in one set of analyses and square root transformed raw values in other sets of analyses? The reader would profit from a bit more detail to what extent Z-transformed values lead to confounding CS+ and CS- values with response magnitude (as indicated in section 4.3.3).

• The authors may want to consult Staib, Castegnetti and Bach 2015 for an investigation of the individual-level z-scoring approach used here.

• Insufficient rationale for analysis: "…if neural correlates to differential SCR were driven more by CS+ or CS-" – What is the motivation for these analyses?

– fMRI

• Similar comments apply (e.g. preprocessing steps in spm, software versions, etc).

• Can the authors provide a more information how exactly the eigenvariates were extracted as there are a number of different ways to do so (different tools, first-level, second level). I also suggest to add a little bit more information/explanation/ discussion what exactly is captured by the eigenvariate that was extracted. Given the level of details provided in the manuscript, I could not completely follow the procedure (i.e., are not 100% sure what was done) and hence interpretation.

• Please clarify whether the CS+ with US were included in the fMRI analyses (or only the CS+ without US)

o Insufficient rationale for some analyses

– Putting together whole-brain and ROI-based data in a regression analysis seems not "fair" (e.g. subject to different biases) to assess the contribution of different brain activations to SCRs

o Additional Analyses – SCR and fMRI: Ideally, data from habituation should be presented/analyzed to make sure that there are no differences between the CS+ and CS- before conditioning.

• Discussion

o At points, the discussion is difficult to follow. I think it needs some cutting and pruning and to be more concise. Some terms are not well defined (e.g, what is "autonomic regulation"?), what are "increases in anxiety"? also, the results from functional connectivity and fMRI studies are combined

o Did the authors record any other outcome measures than SCRs and BOLD fMRI? As the authors only report individual difference analyses with SCRs, the question remains whether results can really be interpreted the way the authors do in the discussion (arousal/salience). It would be very interesting to see comparable analyses with ratings of fear or contingency awareness. If these are not available, I suggest to discuss this point in a bit more detail.

[Editors' note: further revisions were suggested prior to acceptance, as described below.]

Thank you for resubmitting your work entitled "Whole Brain Correlates of Individual Differences in Skin Conductance Responses during Discriminative Fear Conditioning to Social Cues" for further consideration by *eLife*. Your revised article has been evaluated by Drs. Shackman (Reviewing Editor) and Büchel (Senior Editor) and 2 expert reviewers.

Based on a consultation with the reviewers, there is consensus that, while the manuscript has been improved, there are some important remaining issues that need to be addressed.

To summarize:

(1) The Reviewers identified substantial inconsistencies between the point-by-point response and the revised manuscript, making it difficult to judge the revision.

(2) Table 2 is not evident.

(3) Introduction. The Reviewers emphasized the importance of providing a more thorough review of the existing literature and clarifying the specific aims and their rationale. Again, it will be important to adequately address this in both the response letter and the revised manuscript (and to ensure that the two documents are consistent).

(4) The Reviewers raised some concerns with the electrodermal activity (EDA) approach that need to be addressed. Key details are missing. Adequate rationale for the approach should be provided. It may be useful to re-analyze the data using a more optimal approach.

(5) The Reviewers emphasized the need for greater precision in terminology, more accurate descriptions of prior work (e.g. by Fullana and colleagues), and more sober discussion of the results.

(6) The Reviewers underscored the importance of clearly referencing supplementary material (e.g. Appendix) in the main manuscript, to ensure that readers can easily find the referenced information.

(7) The Reviewers highlighted the need to carefully proofread and copy-edit the revision before re-submission to correct any typographic errors.

*Reviewer #1:*

The authors have been responsive to my comments and made substantial changes to their manuscript. Nonetheless, it was quite challenging to review this revision as the changes were not fully transparent. In many places, the quotes included in the point-by-point cover letter did not match the revised text, some new text was not highlighted as new text, deleted text was not shown as deleted text at all (which made it extra difficult for me as a reviewer) and I was unable to identify Table 2 that was newly inserted according to the letter. Despite these issues, the results the authors provide are in principle interesting and the large sample should be noted (even though this was a very specific twin sample) even though effects must be considered small.

Abstract

- "Reproduce" is not the correct term. Replicability is "re-performing the experiment and collecting new data," whereas reproducibility is "re-performing the same analysis with the same code using a different analyst" (Patil et al., 2016). Clearly, the authors did not reproduce any results here. What the authors did here was a generalizability test I assume (also known as conceptual replication).

- Differences between this investigation and previous work need to be carved out more clearly in the introduction and discussion (see also next comment).

- Also please specify if N refers to individuals or pairs of twins.

Introduction

- Prior Work. Provide a less superficial review of the current state of the science. Replication attempts are most useful when it is clearly outlined which effect is aimed to be replicated, a thorough and precise status quo of the literature is provided, and in the case of conceptual replications which procedural and analytical specifications differ from the previous, to-be-replicated work. It would be helpful for the reader if the exact results of previous work are, the employed procedures and analyses of previous work were described and discussed in relation to the present work in more detail.

- Aims. Authors need to clarify the innovation/novelty of the aims and approach. There are inconsistencies between the letter and the manuscript.

- Amygdala Focus. Provide a stronger motivation for the amygdala focus.

"One of the Reviewers noted that, as it is currently written, I found the emphasis on the amygdala problematic. One of the goals of the ms is "to replicate previous findings of an association between individual differences in amygdala response and SCR using an ROI approach. Why is this a goal? Isn't the goal to understand the brain correlates of differences in human conditioning. Maybe the ROI result can be added as an additional result, but it probably should not be part of the goals, at least without much stronger justification in the Introduction. We agree that is problematic, and incorrect, to describe the motivation for focusing on the amygdala as being to 'replicate previous findings'. Our focus on the amygdala is grounded in empirical work in rodents showing that the amygdala is necessary for fear conditioning and theories of the importance of the amygdala for both fear conditioning and SCR modulation in humans. Post hoc, we found evidence for greater responses to CS+ than CS- in the amygdala in our whole-brain voxel-based analysis of fMRI data, suggesting that the amygdala might be involved in the acquisition of conditioned fear in our sample. This reflects the finding of another larger (*n* > 100) neuroimaging study of fear conditioning that reports z-values in the amygdala larger than 5 (Sjouwerman et al., 2020). Therefore, we think a special focus on the amygdala is motivated, and useful, for understanding the regulation of SCR during fear conditioning. However, as the aim was not to replicate the previous findings in the amygdala, we have changed the wording in the last paragraph of the introduction to: 'Also, because the amygdala has been theorized to be important for both fear conditioning and SCR modulation in humans, the association between amygdala response and SCR was assessed using an ROI approach.'"

Comment: If replication was in fact a secondary aim, this needs to be elaborated more in the manuscript. While the authors go into detail in the letter, they only inserted a single sentence on page 3. Please elaborate (also include heterogeneous findings if relevant) and do not change the aims of your work post hoc. It's fine to clarify the aims in response to Reviewer comments, but the aims should not substantively change.

"Another Reviewer noted that Finding out about an SCR-amygdala BOLD correlation is one of the motives of this study. I was left unsure why mass-univariate amygdala activity should correlate with the CS+/CS- difference. According to Fullana et al. (2016), there is no evidence of group-level amygdala activity in the CS+/CS- contrast. On the other hand, patterns of neural responses in the amygdala distinguish CS+/CS- (Bach et al. 2010 J Neurosci, Visser et al. 2011 J Neurosci, Staib and Bach 2018 NIMG). CS+-on neurons are sparse in the amygdala (Reijmers et al. 2007 Science) and is an equal number of CS+ on and CS+ off neurons in the central amygdala (Tovote et al. 2015 Nat Rev Neurosci and original papers referenced therein). On the balance of things, the motivation for looking at amygdala activity in the first place is weak. This needs to be better motivated in light of the available evidence. The motivation to specifically look at the amygdala in relation to SCR comes from the previous work in rodents on threat conditioning as well as neuroimaging studies that have correlated SCR with amygdala responses and shown a positive correlation (see e.g. Labar, Gatenby, Gore, LeDoux and Phelps, 1998; Phelps, Delgado, Nearing and LeDoux et al. 2004; Dunsmoor, Prince, Murty, Kragel and Labar, 2011; Petrovic, Kalisch, Pessiglione, Singer and Dolan, 2008; MacNamara et al., 2015; Marin et al., 2019). We agree that the accumulated evidence for increased amygdala response to CS+ vs CS- is weak as reported by Fullana et al. (2016). However, this does not mean that the amygdala is unimportant in threat conditioning because CS+ on and CS+ off neurons in the amygdala may obscure the signal, as suggested by the reviewer. There could also be multiple other causes for the lack of amygdala findings in the meta-analysis, including varied methodological aspects across studies. Therefore, individual differences in amygdala responses could still be important for understanding SCR. Also, a comparison of amygdala responses to CS+ and CS- in our sample showed that responses were greater in CS+ trials. In the revised version, we acknowledge the lack of strong support for amygdala involvement in neuroimaging studies of conditioning in the introduction. We write: '…However, the involvement of the amygdala in human fear conditioning can be questioned from the results of a meta-analysis of fMRI studies of fear conditioning (Fullana et al., 2016). There are several possible explanations to the lack of aggregated evidence for elevated amygdala responses to the fear cue relative to the control cue. For example, the null result could be an effect of conditioned fear being expressed as a distributed activation pattern across subparts of the amygdala rather than as an increased average amygdala response (Bach et al. 2010 J Neurosci; Reijmers et al. 2007 Science). Even though the evidence for increased amygdala response to CS+ during acquisition remains a topic for discussion, the number of studies that have found a positive correlation between differential SCR and amygdala responses during the acquisition of conditioned fear is substantial (see e.g. Labar, Gatenby, Gore, LeDoux and Phelps, 1998; Phelps, Delgado, Nearing and LeDoux et al. 2004; Dunsmoor, Prince, Murty, Kragel and Labar, 2011; Petrovic, Kalisch, Pessiglione, Singer and Dolan, 2008; MacNamara et al., 2015; Marin et al., 2019), which warrants further investigation of amygdala involvement in SCR regulation in a large sample."

Comment: Note that the text provided here in the letter does not match the text in the manuscript. Please homogenize, and provide adequate detail in the manuscript.

Approach

"SCR/EDA 8 • The peak-scoring windows for the SCR analysis are unclear, and potentially quite problematic. This, together with the comparably large effect size for the CS+/CS- difference in SCR, suggests a potential risk that the authors may have inadvertently looked at outcome-driven (US- or omission-driven) SCR, rather than conditioned SCR. This would call into question the brain-behavior associations. The peak SCR was scored 1 to 4 seconds after the onset of the CS. The CS was presented for 6 seconds, and CS+ presentations co-terminated with a brief electric shock (US). Therefore, the US was presented 2s after the peak SCR was scored, which was enough time to ensure that the US could not have influenced SCRs to the CS+ and the CS-. In the revised version, we also performed a correlation between SCR and fMRI responses when only including the non-reinforced CS+ trials. Results were almost identical to the main analysis including reinforced trials (see Appendix 7)"

Comment: If the authors indeed employed a TTP (Trough To Peak) approach, the 1-4s post-CS refer to the onset of the SCR, not the peak. This approach is uncommon and potentially problematic as they may miss the true-peak which may occur later than 4s post-CS (see e.g. Boucsein 2012, Psychophysiolog). This needs clarification.

"The authors seem to use a 6 s-SOA delay fear conditioning paradigm. SCR scoring was done with ledalab, using the "maximum phasic driver amplitude 1-4 seconds after CS presentation for each participant". The potential problem is the peak detection window. First, can the authors clarify whether the peak window is 1-4 s after CS onset or after CS offset? The time window is after CS onset, not offset. We have clarified this in the methods section under SCR: 'SCR was analyzed using standard peak score (through-to-peak) 1-4 seconds after CS onset for each participant' (p. 11, row 2-3)"

Comment: More information is required. What kind of settings were chosen in Ledalab? What did they use "CDA. Phasicmax" or "TTP. Ampsum" for instance (or yet another option)? The information provided is too little to understand what the authors did.

"Second, do they analyse only non-reinforced trials or also reinforced trials? Both non-reinforced and reinforced trials were analyzed together as SCR was scored prior to US delivery. To ensure that SCR correlation with fMRI responses was equivalent for non-reinforced trials as for all trials, we analyzed these 8 trials separately, as stated earlier. Results were very similar as for all trials. We refer to the new Table 2 in our revised manuscript for statistics."

Comment: I was unable to locate Table 2. Please provide Table 2 or correct the table reference.

"What Ledalab calls the "driver" is a peripheral neural impulse at some unspecified place in the peripheral autonomic system. As can be seen in figure 5a in Benedek and Kaernbach (2010) where SCR was elicited by external events, this "driver" peaks around 2 s after an external event. So, if the US (or US omission) elicits an SCR, then the estimated "driver" will peak 2 s after CS offset and would be included in a 1-4 s window after CS offset. If, on the other hand, there was a gradual increase in SCR > 2 s into the CS, then the driver would peak > 4 s after CS onset and would not be included in a 1-4 s window after CS onset. o In sum, the authors need to better work out and explain their peak scoring windows. They should also compare reinforced and non-reinforced CS+ trials, to rule out any bias in their analysis. Given that Ledalab yields no better results than standard peak scoring, and sometimes worse results (Bach 2014 Biological Psychology), they may want to consider using a standard peak-scoring analysis or similar strategy. (I note that the standard procedures implemented in PsPM – Bach et al. 2020 Beh Res Therapy – are not optimized for this 6-s SOA, even though there is an option that makes the models suitable for this case as well.) We thank the reviewers for this insightful comment. We have described the methodology in a more precise language. After revisiting our analysis, we noted that we had used a standard peak to through method with a time window of 1-4s post-CS onset. See our comment above to Q9."

Comment: While I appreciate the revisions, this is still not clear in the revised manuscript. Please provide a coherent and adequately complete description.

"Insufficient rationale for analysis: "…if neural correlates to differential SCR were driven more by CS+ or CS-" – What is the motivation for these analyses?

Individual differences in SCR difference scores could be associated both with individual differences in SCR to the CS+ and the CS-. Therefore, we wanted to check that SCR to the CS+, and not the CS-, was the reason for the observed correlation between SCR difference scores and fMRI contrast values. We have added this rationale in the results (p 6, row 23)"

Comment: I was unable to locate the rationale on page 6, row 23.

"At points, the authors are insufficiently precise and nuanced in their description of prior work o For instance, please indicate the direction of published findings, rather than just reporting that there was "an association" or "altered responding". We have made changes throughout the manuscript to indicate the direction of associations between SCR and fMRI responses. We have avoided terms like "an association" or "altered responding" to more precisely indicate the directions of findings."

Comment: I was unable to locate the respective changes made (as they were not referenced here and the edits in the manuscript were not fully transparent) and found the reporting oftentimes still too superficial.

"The authors used 4 different trial sequences. Can they provide information on which CS+ trial was the first reinforced trial in these different sequences? The reason I am asking this is that if the first 5 CS+ presentations in sequence#1 were not reinforced but already the first one was reinforced in sequence #2 this would likely lead to differences in learning speed and ultimately average CS discrimination which may impact on the results. Are individual differences in discrimination related to trial sequences? In all sequences, the first CS+ presentation following the 4 CS+ habituation trials was always reinforced. The sequences differed in whether the CS- or CS+ started the acquisition phase. If the reinforced CS+ is always the first trial in the acquisition phase, the CS- trial following the shock will be elevated due to sensitization. This was why the presentation order was counterbalanced. Although it is possible that trial sequences may be related to discrimination, if this is the case, we still show that the individual variation in SCR correlates with fMRI responses irrespective of trial

order."

Comment: Please add this useful information to the manuscript.

"Please clarify in the text whether the amygdala was significantly activated in the whole-brain CS+ vs CS- contrast, as this will be useful for other investigators and future meta-analyses. We now write in the Results section: "We found no differences in neural responses to the CS+ compared to the CS- during habituation. During acquisition, the pattern of activation to the CS+ relative to the CS- was very similar to the pattern reported in the meta-analysis by Fullana et al. (2016) and included large parts of the striatum, the insula, midline areas of the cingulum, lateral temporal cortex, parietal cortex and the supplementary motor areas. Of note, the whole-brain analysis also revealed greater activation to the CS+ than to the CS- in the bilateral amygdala."

Comment: I was confused about this section (page 3 line 45 to end of page) as the authors compare their results to those of Fullana who looked at the CS+/CS- contrast in fMRI but not at a correlation with SCRs. I think this is also done in other sections of the manuscript. It needs to be made clear in the manuscript that Fullana did not investigate brain-behavior associations (i.e. neural correlates of differential SCRs and how the results relate to each other).

"Discussion 41 o Authors note that individual differences in SCRs are stable and provide 3 references for this. They may want to double-check if these references really show demonstrate the stability of individual differences in CS discrimination. If I am not mistaken, neither Fredriksson (1993) nor Zeidan (2012) report stability measures for CS discrimination per se (but only for CS+ and CS- individually). This is a good point. We now refer to Fredrikson (1993) and Zeidan (2012) in terms that they have shown that SCR during fear conditioning is relatively reproducible. We do not refer to CS differences here. (First sentence in the discussion, p. 7, row 5)"

Comment: Technically, what was studied by Fredrikssion and Zeidan was reliability, not reproducibility (see above for a definition of what reproducibility refers to). I suggest being more precise here. Also, I do not think it becomes clear from the revision that the findings by Zeidan, Torrents-Rodas and Fredriksson do not refer to CS discrimination (or do they?). On the contrary, from the wording the authors have chosen, I would infer that this is about CS discrimination. As this work is mainly about CS discrimination this is important. I refer the authors to other work that also investigated fMRI test-retest reliability and/or test-retest reliability for CS discrimination and once more would appreciate (again) more precision in reporting.

Published

Ridderbusch, I. C., Wroblewski, A., Yang, Y., Richter, J., Hollandt, M., Hamm, A. O.,.…

Straube, B. (2021). Neural adaptation of cingulate and insular activity during delayed fear extinction: A replicable pattern across assessment sites and repeated measurements.

NeuroImage, 237, 118157. https://doi.org/10.1016/j.neuroimage.2021.118157

Pre-prints

Samuel E Cooper, Joseph E Dunsmoor, Kathleen Koval, Emma Pino, Shari Steinman, Test-Retest Reliability of Human Threat Conditioning and Generalization , PsyArXiv, https://psyarxiv.com/84uqz/

Maren Klingelhöfer-Jens, Mana R. Ehlers, Manuel Kuhn, Vincent Keyaniyan, Tina B. Lonsdorf. Robust group- but limited individual-level (longitudinal) reliability and insights into cross-phases response prediction of conditioned fear doi: https://doi.org/10.1101/2022.03.15.484434

---

## [Author Response]

Essential Revisions• Authors need to better integrate genetic relatedness into their report.o The study sample is relatively large (*N* = 285) – yet the sample is special in that participants were genetically related and as siblings (twins, mono- and dizygotic twins) shared environmental influences. This become only apparent in the method section (and discussion?) but needs to be mentioned upfront in the abstract, intro and included in the discussion as this may have an impact on the results. From the methods section it remained unclear how many pairs of di- and monozyotic twins were included in the study and more information on the sample (age range for instance) would be desirable.

We now mention in the abstract and last paragraph of the introduction that the sample consists of twins. The number of MZ and DZ pairs is also stated in the methods section (first paragraph). We checked whether the same regions were correlated with SCR when only one twin of each pair was included and showed that this was the case. The p-values were slightly lower because of the halved sample size, but the correlation coefficients remained. We discuss the eventual impact of the sample being twins in the limitation section of the discussion (p.8, lines 54-58 and p. 9 lines 1-14).

The age range is now mentioned in the methods section. Thanks for noticing that this information was lacking.

• Polish the manuscript. There are grammatical errors. Wording and clarity could be improved. If the manuscript is meant for neuroscientists and psychologists in general (not only human fear conditioning experts), the reader probably needs some more background on some of the topics dealt with.

We have changed the first paragraph. We now include more background to why fear conditioning is an important model for psychiatric disorders. We also refer to studies showing that autonomic activity is elevated in many psychiatric disorders. The neural correlates of autonomic activity (SCR) is therefore an important subject for understanding physical disease (cardiovascular, stroke) leading to early death in psychiatric disorders. We this this more general topic is of interest to the broad readership of *eLife*.

• The authors need to be more precise and nuanced in their description of prior work (see below)

We have included several new references to recent work on fear conditioning in psychiatric disorders including schizophrenia (Tuominen et al., 2021) and OCD (Cooper and Dunsmoor, 2021). We have also included references to new studies on the neural correlates of conditioned SCR (Savage et al., 2021).

• Introductiono Provide a less superficial review of the current state of the science. Replication attempts are most useful when it is clearly outlined which effect is aimed to be replicated, a thorough and precise status quo of the literature is provided and in case of conceptual replications which procedural and analytical specifications differ from the previous, to-be-replicated work. It would be helpful for the reader if the exact results of previous work are, the employed procedures and analyses of previous work were described and discussed in relation to the present work in more detail.

The aim of the study was not to replicate previous results, but to find whole brain correlates of SCR to better understand neural regulation of autonomic activity. We hope that we have made this clearer in the introduction. We describe the need for whole brain analyses of correlations to SCR in the next to last paragraph (p. 2-3). See also our response to point 4 and 5 below.

o Authors need to better clarify the innovation/novelty of the aims and approach

This is a great point raised by the reviewers and we have emphasized the novelty in the introduction of the revised version. The novelty of our study is the use of a whole brain approach to investigate the neural correlates of autonomic conditioning, which calls for an unprecedented sample size. We have made this clearer in the next to last paragraph of the introduction. We now say that: ‘In order to better understand the brain correlates of differences in human conditioning, it seems necessary to calculate correlations across the whole brain in a far larger sample than has previously been used. The use of a whole brain approach in establishing neural correlates of individual differences in conditioned SCR has potential to identify novel brain regions influencing fear conditioning and psychopathology.’

o Provide a stronger motivation for the amygdala focus– One of the Reviewers noted that, as it is currently written, I found the emphasis on the amygdala problematic. One of the goals of the manuscript is "to replicate previous findings of an association between individual differences in amygdala response and SCR using an ROI approach". Why is this a goal? Isn't the goal to understand the brain correlates of differences in human conditioning. Maybe the ROI result can be added as an additional result, but it probably should not be part of the goals, at least without much stronger justification in the Introduction.

We agree that is problematic, and incorrect, to describe the motivation for focusing on the amygdala as being to ‘replicate previous findings’. Our focus on the amygdala is grounded in empirical work in rodents showing that the amygdala is necessary for fear conditioning and theories of the importance of the amygdala for both fear conditioning and SCR modulation in humans. Post hoc, we found evidence for greater responses to CS+ than CS- in the amygdala in our whole brain voxel-based analysis of fMRI data, suggesting that the amygdala might be involved in the acquisition of conditioned fear in our sample. This reflects the finding of another larger (*n* > 100) neuroimaging study of fear conditioning that report z-values in the amygdala larger than 5 (Sjouwerman et al., 2020). Therefore, we think a special focus on the amygdala is motivated, and useful, for understanding the regulation of SCR during fear conditioning.

However, as the aim was not to replicate previous finding in the amygdala, we have changed the wording in the last paragraph of the introduction to: ‘Also, because the amygdala has been theorized to be important for both fear conditioning and SCR modulation in humans, the association between amygdala response and SCR was assessed using an ROI approach.’

– Another Reviewer noted that, Finding out about an SCR-amygdala BOLD correlation is one of the motives of this study. I was left unsure why mass-univariate amygala activity should correlate with the CS+/CS- difference. According to Fullana et al. (2016), there is no evidence of group-level amygdala activity in the CS+/CS- contrast. On the other hand, patterns of neural responses in the amygdala distinguish CS+/CS- (Bach et al. 2010 J Neurosci, Visser et al. 2011 J Neurosci, Staib and Bach 2018 NIMG). CS+-on neurons are sparse in the amygdala (Reijmers et al. 2007 Science) and is an equal number of CS+ on and CS+ off neurons in the central amygdala (Tovote et al. 2015 Nat Rev Neurosci and original papers referenced therein). On the balance of things, the motivation for looking at amygdala activity in the first place is weak. This needs to be better motivated in light of the available evidence

The motivation to specifically look at the amygdala in relation to SCR comes from the previous work in rodents on threat conditioning as well as neuroimaging studies that have correlated SCR with amygdala responses and shown a positive correlation (see e.g. Labar, Gatenby, Gore, LeDoux and Phelps, 1998**;** Phelps, Delgado, Nearing and LeDoux et al. 2004**;** Dunsmoor, Prince,Murty, Kragel and Labar, 2011; Petrovic, Kalisch, Pessiglione, Singer and Dolan, 2008**;** MacNamara et al., 2015; Marin et al., 2019). We agree that the accumulated evidence for increased amygdala response to CS+ vs CS- is weak as reported by Fullana et al. (2016). However, this does not mean that the amygdala is unimportant in threat conditioning because CS+ on and CS+ off neurons in the amygdala may obscure the signal, as suggested by the reviewer. There could also be multiple other causes to the lack of amygdala findings in the meta-analysis, included varied methodological aspects across studies. Therefore, individual differences in amygdala responses could still be important for understanding SCR. Also, a comparison of amygdala responses to CS+ and CS- in our sample showed that responses were greater to CS+ trials.

In the revised version, we acknowledge the lack of strong support for amygdala involvement in neuroimaging studies of conditioning in the introduction. We write:

‘…However, the involvement of the amygdala in human fear conditioning can be questioned from the results of a meta-analysis of fMRI studies of fear conditioning (Fullana et al., 2016). There are several possible explanations to the lack of aggregated evidence for an elevated amygdala responses to the fear cue relative to the control cue. For example, the null result could be an effect of conditioned fear being expressed as a distributed activation pattern across subparts of the amygdala rather than as an increased average amygdala response (Bach et al. 2010 J Neurosci; Reijmers et al. 2007 Science). Even though the evidence for increased amygdala response to CS+ during acquisition remains a topic for discussion, the number of studies that have found a positive correlation between differential SCR and amygdala responses during acquisition of conditioned fear is substantial (see e.g. Labar, Gatenby, Gore, LeDoux and Phelps, 1998**;** Phelps, Delgado, Nearing and LeDoux et al. 2004**;** Dunsmoor, Prince, Murty, Kragel and Labar, 2011; Petrovic, Kalisch, Pessiglione, Singer and Dolan, 2008**;** MacNamara et al., 2015; Marin et al., 2019), which warrants further investigation of amygdala involvement in SCR regulation in a large sample.’

• Approacho Authors need to clarify the approach and provide some crucial missing details that have the potential to markedly influence the results and conclusions– Enrollment criteria• Some inclusion/exclusion criteria are not well defined (e.g., "current alcohol or drug-related problems") or unclear (e.g., why should someone receiving psychological treatment be excluded? were only psychotropic medications -and not other medications- excluded? )

We have rephrased the exclusion criteria to make them more precise. We changed ‘current alcohol or drug related problems’ to ‘ongoing substance abuse’ (p. 9, row 39). We now also specify that ‘Non-psychotropic medication was not an exclusion criterion.’ (p. 9, row 40)

Participants receiving psychological treatment at the time of the assessment were excluded as treatment could have an effect on brain responses to emotional stimuli. Although this may be less of a problem in the current analysis of brain correlates of SCR, it could be a problem when analyzing correlations between individuals in a twin pair if one individual receives treatment and one does not receive treatment (these results will be reported elsewhere). This is why we chose to include this exclusion criterion.

– SCR/EDA• The peak-scoring windows for the SCR analysis are unclear, and potentially quite problematic. This, together with the comparably large effect size for the CS+/CS- difference in SCR, suggests a potential risk that the authors may have inadvertently looked at outcome-driven (US- or omission-driven) SCR, rather than conditioned SCR. This would call into question the brain-behavior associations

The peak SCR was scored 1 to 4 seconds after the onset of the CS. The CS was presented for 6 seconds, and CS+ presentations co-terminated with a brief electric shock (US). Therefore, the US was presented 2s after the peak SCR was scored, which was enough time to ensure that the US could not have influenced SCRs to the CS+ and the CS-. In the revised version, we also performed a correlation between SCR and fMRI responses when only including the non-reinforced CS+ trials. Results were almost identical to the main analysis including reinforced trials (see Appendix 7)

o The authors seem to use a 6 s-SOA delay fear conditioning paradigm. SCR scoring was done with ledalab, using the "maximum phasic driver amplitude 1-4 seconds after CS presentation for each participant". The potential problem is the peak detection window. First, can the authors clarify whether the peak window is 1-4 s after CS onset or after CS offset?

The time window is after CS onset, not offset. We have clarified this in the methods section under SCR: ‘SCR was analyzed using standard peak score (through-to-peak) 1-4 seconds after CS onset for each participant’ (p. 11, row 2-3)

Second, do they analyse only non-reinforced trials or also reinforced trials?

Both non-reinforced and reinforced trials were analyzed together as SCR was scored prior to US delivery. To ensure that SCR correlation with fMRI responses was equivalent for nonreiforced trials as for all trials, we analyzed these 8 trials separately, as stated earlier. Results were very similar as for all trials. We refer to the new Table 2 in our revised manuscript for statistics.

o What Ledalab calls the "driver" is a peripheral neural impulse at some unspecified place in the peripheral autonomic system. As can be seen in figure 5a in Benedek and Kaernbach (2010) where SCR were elicited by external events, this "driver" peaks around 2 s after an external event. So, if the US (or US omission) elicits an SCR, then the estimated "driver" will peak 2 s after CS offset and would be included in a 1-4 s window after CS offset. If, on the other hand, there was a gradual increase in SCR > 2 s into the CS, then the driver would peak > 4 s after CS onset and would not be included in a 1-4 s window after CS onset.o In sum, the authors need to better work out and explain their peak scoring windows. They should also compare reinforced and non-reinforced CS+ trials, to rule out any bias in their analysis. Given that Ledalab yields no better results than standard peak scoring, and sometimes worse results (Bach 2014 Biological Psychology), they may want to consider using a standard peak-scoring analysis or similar strategy. (I note that the standard procedures implemented in PsPM – Bach et al. 2020 Beh Res Therapy – are not optimized for this 6-s SOA, even though there is an option that makes the models suitable for this case as well.)

We thank the reviewers for this insightful comment. We have described the methodology in a more precise language. After revisiting our analysis, we noted that we had used a standard peak to through method with a time window of 1-4s post CS onset. See our comment above to Q9.

• In general, the authors need to provide more details on the SCR data acquisition, processing, and analyses (e.g. the versions of the software used and specific settings/parameters e.g., sampling rate, all filters, downsampling if any, derivative scores, quality assurance/control procedures, etc.).

Thanks for making us aware of the lack of detail. We now provide software version, sampling rate, hardware filters, downsampling and quality control. On p. 10 row 29 we write: “The signal was high-pass (0.05Hz) filtered using the built in BIOPAC hardware Butterworth filter. SCRs were scored using Ledalab software package (v 3.4.9) (Benedek and Kaernbach, 2010) implemented in Matlab (Mathworks, Inc, Natick, MA). Minimum response threshold was set to 0.01 µS. After filtering and before the analysis, the SCR signal was downsampled from 2000Hz to 10Hz. SCR was analyzed using standard peak score (through-to-peak) 1-4 seconds after CS onset for each participant.”

• The pre-processing (filtering) is not sufficiently justified. Authors may want to consider the results by Privratsky et al. 2020 (https://pubmed.ncbi.nlm.nih.gov/33075428/) to guide their choice of filters

We thank the reviewers for pointing us to this interesting paper on filtering of the skin conductance signal. As stated in our response to Q10, filtering of skin conductance was performed using the 0.05Hz high-pass Butterworth filter in the BIOPAC measurement system.

• The authors need to elaborate on the advantages of using Z-transformed SCR in one set of analyses and square root transformed raw values in other sets of analyses? The reader would profit from a bit more detail to what extent Z-transformed values lead to confounding CS+ and CS- values with response magnitude (as indicated in section 4.3.3).

We have provided an explanation in the methods section as to why we used Z-transformed SCR in one set of analysis and square root transformed raw values in the other (p 13, row 18).

“Specifically, since the Z transformed SCR is defined as the difference between the SCR on a given trial (or trial type) and the average SCR across all trials, divided by the standard deviation of SCRs across all trials, this means that using Z scores conflates CS+ and CS- responding with responses to both average CS+ and CS- inclusively, and therefore cannot be used to determine the influence of neural activity on a specific trial type”

• The authors may want to consult Staib, Castegnetti and Bach 2015 for an investigation of the individual-level z-scoring approach used here.

We thank the reviewers for pointing us to this interesting paper. We now refer to this study when describing the z-scoring approach.

• Insufficient rationale for analysis: "…if neural correlates to differential SCR were driven more by CS+ or CS-" – What is the motivation for these analyses?

Individual differences in SCR difference scores could be associated both with individual differences in SCR to the CS+ and the CS-. Therefore, we wanted to check that SCR to the CS+, and not the CS-, was the reason for the observed correlation between SCR difference scores and fMRI contrast values. We have added this rationale in the results (p 6, row 23)

– fMRI• Similar comments apply (e.g. preprocessing steps in spm, software versions, etc).

We now provide more details in the description of the preprocessing steps in SPM including version, co-registration of functional images to the anatomical image, as well as segmentation and warping to MNI space. (p 12 rows 11-19)

• Can the authors provide a more information how exactly the eigenvariates were extracted as there are a number of different ways to do so (different tools, first-level, second level). I also suggest to add a little bit more information/explanation/ discussion what exactly is captured by the eigenvariate that was extracted. Given the level of details provided in the manuscript, I could not completely follow the procedure (i.e., are not 100% sure what was done) and hence interpretation.

We extracted the eigenvariates of every significant cluster of voxels in the whole brain analysis of individual differences in SCR. The result is one value per cluster for each individual. The eigenvariates are based on the contrast image from the first level contrast CS+ > CS- and are strongly correlated (r >.9) to the mean of the same voxels. The eigenvariates are used instead of the mean as they are less sensitive to extreme values in individual voxels. Eigenvariates are extracted using a Singular Value Decomposition (SVD) of contrast values in each cluster from the first level contrast within SPM12. We have added information on the extraction process in the methods section (p. 13 rows 9-17).

• Please clarify whether the CS+ with US were included in the fMRI analyses (or only the CS+ without US)

Both reinforced and non-reinforced CS+ trials were used in the fMRI analysis. Following the suggestion of the reviewers, we also tested whether results came out different if only CS+ without US trials were used and found that the results were almost identical. However, because fewer trials were used (8 instead of 16) statistical power was smaller. To increase power we therefore report the results when all trials were used but also give results from the correlation with non-reinforced CS+ trials in Appendix 7.

o Insufficient rationale for some analyses– Putting together whole-brain and ROI-based data in a regression analysis seems not "fair" (e.g. subject to different biases) to assess the contribution of different brain activations to SCRs

We agree. Because our focus is on whole-brain results and not on amygdala results, we chose to let go of the analyses that included the amygdala. The analysis without the amygdala suggests that all regions contribute to SCR, with no region sticking out as driving SCR more strongly than any other. This may suggest that the regions associated with SCR form part of a network related to SCR, which we now discuss in the Discussion section of the revised manuscript. (paragraph 3 in Discussion p. 7).

o Additional Analyses – SCR and fMRI: Ideally, data from habituation should be presented/analyzed to make sure that there are no differences between the CS+ and CS- before conditioning.

We now present the analysis of SCR and fMRI data from the habituation phase. There was no difference in SCR to CS+ and CS- (p. 3 row 23). There were no differences in fMRI responses to CS+ relative to CS- which we report (p. 3 row 48).

• Discussiono At points, the discussion is difficult to follow. I think it needs some cutting and pruning and to be more concise. Some terms are not well defined (e.g, what is "autonomic regulation"?), what are "increases in anxiety"? also, the results from functional connectivity and fMRI studies are combined

We have edited the discussion following the suggestion from the reviewers to make it more concise. Following a summary, we now first discuss the overlap between the whole brain correlations with SCR and the regions that showed increased response to the CS+ vs. CS- in the meta-analysis by Fullana et al. We next discuss the results from the hierarchical regression analysis which suggest that the whole brain SCR correlates may form part of a single network, as no region sticks out as explaining a separate portion of the variance in SCR when compared with the other regions. We think that this network overlaps well with the midcingulo-insula network previously described by Uddin and others. Lastly, we discuss implications for psychiatric and physical health.

We have changed the term “autonomic regulation” to regulation of autonomic activity. We have also changed “increases in anxiety” and now discuss anxiety disorders.

o Did the authors record any other outcome measures than SCRs and BOLD fMRI? As the authors only report individual difference analyses with SCRs, the question remains whether results can really be interpreted the way the authors do in the discussion (arousal/salience). It would be very interesting to see comparable analyses with ratings of fear or contingency awareness. If these are not available, I suggest to discuss this point in a bit more detail.

We recorded button presses during the fMRI-task. Participants pressed one of two buttons each time a CS was displayed to indicate whether they would receive an electric shock (coded 1) or not (coded 0). The mean response was computed for the CS+ and the CS- presentations. This information has been added to the methods section (p. 12 rows 12-15).

“The average shock expectancy was greater to the CS+ (*M* = 0.68, *SD* = 0.26) than the CS- (*M* = 0.09, *SD* = 0.21) (*t*_*285*_ = 25.52; *p* < 0.05), indicating that participants learned the contingency.”

This information has been added to the Results section (p. 3 rows 28-30).

We controlled for shock expectancy in the whole brain analysis. As can be seen in Appendix 7, this had only a marginal effect on t-values in the peak voxels from the main analysis.

[Editors' note: further revisions were suggested prior to acceptance, as described below.]

To summarize:(1) The Reviewers identified substantial inconsistencies between the point-by-point response and the revised manuscript, making it difficult to judge the revision.

We have thoroughly been checking that the responses to reviewers correspond to the changes made to the manuscript in this revised version. We apologize for the previous inconsistencies which we understand must have made the reviewers’ jobs frustrating.

(2) Table 2 is not evident.

We have moved the former table 2 showing the sensitivity analysis to Appendix 7. We also have rewritten the text explaining what is tabulated in the table. Also, we have provided an additional Appendix 8 detailing the methods used for one of the sensitivity analyses reported in Appendix 7 (DCM-based analysis in PsPM).

(3) Introduction. The Reviewers emphasized the importance of providing a more thorough review of the existing literature and clarifying the specific aims and their rationale. Again, it will be important to adequately address this in both the response letter and the revised manuscript (and to ensure that the two documents are consistent).

We have added several references and have tried to make the description of the previous studies of the brain correlates of SCR more comprehensive. To this end we have added a table (table 1) that summarizes results and analyses methods used in these studies.

(4) The Reviewers raised some concerns with the electrodermal activity (EDA) approach that need to be addressed. Key details are missing. Adequate rationale for the approach should be provided. It may be useful to re-analyze the data using a more optimal approach.

We have added details regarding the SCR analysis. We have also used new analysis approaches following suggestions from reviewers that we have correlated with brain data. Correlation results are consistent across these different choices of SCR analysis. Hence, results do not seem to be sensitive to the particular type of SCR analysis used.

(5) The Reviewers emphasized the need for greater precision in terminology, more accurate descriptions of prior work (e.g. by Fullana and colleagues), and more sober discussion of the results.

We have addressed this point in the revised manuscript and have deleted parts of the discussion that reviewers thought was a little too speculative.

(6) The Reviewers underscored the importance of clearly referencing supplementary material (e.g. Appendix) in the main manuscript, to ensure that readers can easily find the referenced information.

We apologize for having missed some references to the supplemental material in the previous revision of the manuscript. This has been corrected in the present version so that all supplemental material is referenced in the text. Please note that the supplemental material is now referenced as ‘Appendix’ in line with eLifes’ rules on naming conventions.

(7) The Reviewers highlighted the need to carefully proofread and copy-edit the revision before re-submission to correct any typographic errors.

We have used a native English-speaking proofreader to correct typos and grammatical errors. We hope the manuscripts reads better now.

Reviewer #1:The authors have been responsive to my comments and made substantial changes to their manuscript. Nonetheless, it was quite challenging to review this revision as the changes were not fully transparent. In many places, the quotes included in the point-by-point cover letter did not match the revised text, some new text was not highlighted as new text, deleted text was not shown as deleted text at all (which made it extra difficult for me as a reviewer) and I was unable to identify Table 2 that was newly inserted according to the letter. Despite these issues, the results the authors provide are in principle interesting and the large sample should be noted (even though this was a very specific twin sample) even though effects must be considered small.

We apologize for the inconsistences in the manuscript. We have now carefully reviewed the changes to minimize errors and provide Related Manuscript files containing properly tracked changes between our revisions. The table we previously referred to as Table 2 is available in Appendix 7.

Abstract- "Reproduce" is not the correct term. Replicability is "re-performing the experiment and collecting new data," whereas reproducibility is "re-performing the same analysis with the same code using a different analyst" (Patil et al., 2016). Clearly, the authors did not reproduce any results here. What the authors did here was a generalizability test I assume (also known as conceptual replication).

We thank the reviewer for pointing this out and have throughout our manuscript changed the term ‘reproduce’ to other terms that are more appropriate:

Abstract, p. 1, Line 19-20: A ROI analysis additionally showed a positive correlation between amygdala activity and conditioned SCR in line with previous reports. Introduction, p. 5 Line 25-29: Based on findings from previous studies of individual differences in conditioned SCR (Labar, Gatenby, Gore, LeDoux and Phelps, 1998; Phelps, Delgado, Nearing and LeDoux et al. 2004; Dunsmoor, Prince, Murty, Kragel and Labar, 2011; Petrovic, Kalisch, Pessiglione, Singer and Dolan, 2008; MacNamara et al., 2015; Marin et al., 2019), we also hypothesized a positive correlation between SCR and amygdala activation. Discussion, p. 10, Line 13-14: “As expected, we found a correlation between individual differences in conditioned SCR and amygdala activity in line with previous reports…” Materials and methods, P. 17 Line 21-22: “Secondly, we tested a hypothesized association between individual differences in amygdala response and SCR using a regionof-interest (ROI) analysis.”

- Differences between this investigation and previous work need to be carved out more clearly in the introduction and discussion (see also next comment).

The main differences between our work and previous studies is the low number of participants in previous studies and a priori definitions of brain regions (p. 2, rows 40-51, continued on p.3, rows 1-6; p. 5, rows 1-11). We summarize characteristics of a representative selection of previous studies in Table 1. See also our response to “Prior work” below for full summary.

- Also please specify if N refers to individuals or pairs of twins.

We have now specified in the abstract and throughout the manuscript that N refers to individuals (p. 1, row 16, p. 5, row 15 and 19).

Introduction- Prior Work. Provide a less superficial review of the current state of the science. Replication attempts are most useful when it is clearly outlined which effect is aimed to be replicated, a thorough and precise status quo of the literature is provided, and in the case of conceptual replications which procedural and analytical specifications differ from the previous, to-be-replicated work. It would be helpful for the reader if the exact results of previous work are, the employed procedures and analyses of previous work were described and discussed in relation to the present work in more detail.

We have now more precisely clarified the differences between previous studies and our work with regards to analysis and procedure. See p. 2, rows 8-51; p. 3, rows 1-6, and Table 1.

- Aims. Authors need to clarify the innovation/novelty of the aims and approach. There are inconsistencies between the letter and the manuscript.

We have now made sure to refer to all changes in the manuscript in the response letter. Regarding novelty, we compare our study to previous studies and write: “… previous studies only studied correlations to SCR in a handful of brain regions and in relatively small samples” (p. 5, 12-13). Regarding aims, we write: “the primary aim of this study was to investigate the whole brain correlations of individual differences in conditioned SCR by analyzing data from a large twin sample performing a fear conditioning task ” (p. 5, rows 13-15). Then we write in our second aim based on previous results, but in small sample sizes: “we hypothesized a positive correlation between SCR and amygdala activation.” (p. 5, rows 28-29). The novelty regarding the second aim was the increased number of participants in our study as compared to previous studies. Our last aim was: “a last aim of the present study was to determine whether areas whose activity explained significant individual variation in conditioned SCR did so independently of one other”. (p. 5, rows 31-32). The last aim is novel and has to our knowledge not been evaluated before. We now hope that the novelty of the study is clear and that our aims are clearly stated.

- Amygdala Focus. Provide a stronger motivation for the amygdala focus."One of the Reviewers noted that, as it is currently written, I found the emphasis on the amygdala problematic. One of the goals of the ms is "to replicate previous findings of an association between individual differences in amygdala response and SCR using an ROI approach. Why is this a goal? Isn't the goal to understand the brain correlates of differences in human conditioning. Maybe the ROI result can be added as an additional result, but it probably should not be part of the goals, at least without much stronger justification in the Introduction. We agree that is problematic, and incorrect, to describe the motivation for focusing on the amygdala as being to 'replicate previous findings'. Our focus on the amygdala is grounded in empirical work in rodents showing that the amygdala is necessary for fear conditioning and theories of the importance of the amygdala for both fear conditioning and SCR modulation in humans. Post hoc, we found evidence for greater responses to CS+ than CS- in the amygdala in our whole-brain voxel-based analysis of fMRI data, suggesting that the amygdala might be involved in the acquisition of conditioned fear in our sample. This reflects the finding of another larger (*n* > 100) neuroimaging study of fear conditioning that reports z-values in the amygdala larger than 5 (Sjouwerman et al., 2020). Therefore, we think a special focus on the amygdala is motivated, and useful, for understanding the regulation of SCR during fear conditioning. However, as the aim was not to replicate the previous findings in the amygdala, we have changed the wording in the last paragraph of the introduction to: 'Also, because the amygdala has been theorized to be important for both fear conditioning and SCR modulation in humans, the association between amygdala response and SCR was assessed using an ROI approach.'"Comment: If replication was in fact a secondary aim, this needs to be elaborated more in the manuscript. While the authors go into detail in the letter, they only inserted a single sentence on page 3. Please elaborate (also include heterogeneous findings if relevant) and do not change the aims of your work post hoc. It's fine to clarify the aims in response to Reviewer comments, but the aims should not substantively change.

We have more clearly motivated the aim to evaluate the correlation between SCR and amygdala responses during fear conditioning. We motivate this aim in the introduction p. 2 rows 8-39 where we cite two large, recent, fMRI studies that have found amygdala activation during fear conditioning (Kastrati et al., 2022; Wen et al., 2022). Also see Table 1 for an overview of previous study findings. Please, see also our response to the next question.

"Another Reviewer noted that Finding out about an SCR-amygdala BOLD correlation is one of the motives of this study. I was left unsure why mass-univariate amygdala activity should correlate with the CS+/CS- difference. According to Fullana et al. (2016), there is no evidence of group-level amygdala activity in the CS+/CS- contrast. On the other hand, patterns of neural responses in the amygdala distinguish CS+/CS- (Bach et al. 2010 J Neurosci, Visser et al. 2011 J Neurosci, Staib and Bach 2018 NIMG). CS+-on neurons are sparse in the amygdala (Reijmers et al. 2007 Science) and is an equal number of CS+ on and CS+ off neurons in the central amygdala (Tovote et al. 2015 Nat Rev Neurosci and original papers referenced therein). On the balance of things, the motivation for looking at amygdala activity in the first place is weak. This needs to be better motivated in light of the available evidence. The motivation to specifically look at the amygdala in relation to SCR comes from the previous work in rodents on threat conditioning as well as neuroimaging studies that have correlated SCR with amygdala responses and shown a positive correlation (see e.g. Labar, Gatenby, Gore, LeDoux and Phelps, 1998; Phelps, Delgado, Nearing and LeDoux et al. 2004; Dunsmoor, Prince, Murty, Kragel and Labar, 2011; Petrovic, Kalisch, Pessiglione, Singer and Dolan, 2008; MacNamara et al., 2015; Marin et al., 2019). We agree that the accumulated evidence for increased amygdala response to CS+ vs CS- is weak as reported by Fullana et al. (2016). However, this does not mean that the amygdala is unimportant in threat conditioning because CS+ on and CS+ off neurons in the amygdala may obscure the signal, as suggested by the reviewer. There could also be multiple other causes for the lack of amygdala findings in the meta-analysis, including varied methodological aspects across studies. Therefore, individual differences in amygdala responses could still be important for understanding SCR. Also, a comparison of amygdala responses to CS+ and CS- in our sample showed that responses were greater in CS+ trials. In the revised version, we acknowledge the lack of strong support for amygdala involvement in neuroimaging studies of conditioning in the introduction. We write: '…However, the involvement of the amygdala in human fear conditioning can be questioned from the results of a meta-analysis of fMRI studies of fear conditioning (Fullana et al., 2016). There are several possible explanations to the lack of aggregated evidence for elevated amygdala responses to the fear cue relative to the control cue. For example, the null result could be an effect of conditioned fear being expressed as a distributed activation pattern across subparts of the amygdala rather than as an increased average amygdala response (Bach et al. 2010 J Neurosci; Reijmers et al. 2007 Science). Even though the evidence for increased amygdala response to CS+ during acquisition remains a topic for discussion, the number of studies that have found a positive correlation between differential SCR and amygdala responses during the acquisition of conditioned fear is substantial (see e.g. Labar, Gatenby, Gore, LeDoux and Phelps, 1998; Phelps, Delgado, Nearing and LeDoux et al. 2004; Dunsmoor, Prince, Murty, Kragel and Labar, 2011; Petrovic, Kalisch, Pessiglione, Singer and Dolan, 2008; MacNamara et al., 2015; Marin et al., 2019), which warrants further investigation of amygdala involvement in SCR regulation in a large sample."Comment: Note that the text provided here in the letter does not match the text in the manuscript. Please homogenize, and provide adequate detail in the manuscript.

We apologize for this discrepancy and have now exactly matched the text in the manuscript and our response here. We now motivate our focus on the amygdala the following way in the manuscript. p. 2 rows 8-39:

“Previous studies of the neural correlates of individual differences in conditioned SCR, generally defined as the difference in average SCR score between CS+ and CS- presentations during acquisition (see Lonsdorf and Merz, 2017, for a discussion of definitions), have focused on either one or a few brain regions using region of interest analyses. Many of these studies have found positive correlations with neural responses in the amygdala (Labar, Gatenby, Gore, LeDoux and Phelps, 1998; Phelps, Delgado, Nearing and LeDoux et al. 2004; Dunsmoor, Prince, Murty, Kragel and Labar, 2011; Petrovic, Kalisch, Pessiglione, Singer and Dolan, 2008; MacNamara et al., 2015; Marin et al., 2019). Neuroimaging studies of within-subject variation in conditioned SCR have also generally found positive correlations to amygdala responses (Cheng, Knight, Smith, Stein and Helmstetter, 2003; Knight, Nguyen, and Bandettini, 2005; Cheng, Knight, Smith and Helmstetter, 2006), although exceptions exist (Sjouwerman, Sharfenort and Lonsdorf, 2020; Savage et al., 2021). The findings from studies that report a positive relationship between SCR and amygdala activity are in line with the general understanding of fear conditioning from animal models, where a neural circuitry centered on the amygdala is responsible for the acquisition of conditioned fear responses (LeDoux 2000; Davis 2000). They also complement those human lesion studies demonstrating either diminished or absent conditioned SCR following amygdala damage (Labar et al. 1995; Bechara et al., 1995), although not all studies have found such an effect (Åhs et al., 2010; for a review see Ojala and Bach, 2020). Further, the involvement of the amygdala in human fear conditioning has been questioned based on the results of a meta-analysis of fMRI studies investigating fear conditioning (Fullana et al., 2016) and based on studies showing unexpected, increased amygdala responses to the CS- compared to the CS+ (see e.g. Visser et al. 2021). Such results could arise from distributed representations of the CS+ and CS- in the amygdala (Bach, Weiskopf and Dolan, 2011; Reijmers, Perkins, Matsuo and Mayford 2007) or from a need for larger sample sizes to detect differential responses in the amygdala. Speaking to the latter idea, two independent studies, each including hundreds of participants, have recently reported increased CS+, relative to CS-, activation in the amygdala (Kastrati et al., 2022; Wen et al., 2022). Amygdala activation to the CS+ was primarily detected during the first trials of acquisition, whereas CS- activity was larger in the end of acquisition (Wen et al., 2022). The results of these large studies, together with the fairly consistent findings of correlated individual differences in conditioned SCR and amygdala activation (Labar, Gatenby, Gore, LeDoux and Phelps, 1998; Phelps, Delgado, Nearing and LeDoux et al. 2004; Dunsmoor, Prince, Murty, Kragel and Labar, 2011; Petrovic, Kalisch, Pessiglione, Singer and Dolan, 2008; MacNamara et al., 2015; Marin et al., 2019), support our hypothesis that amygdala activation should be positively correlated with SCR in this study.”

Approach"SCR/EDA 8 • The peak-scoring windows for the SCR analysis are unclear, and potentially quite problematic. This, together with the comparably large effect size for the CS+/CS- difference in SCR, suggests a potential risk that the authors may have inadvertently looked at outcome-driven (US- or omission-driven) SCR, rather than conditioned SCR. This would call into question the brain-behavior associations. The peak SCR was scored 1 to 4 seconds after the onset of the CS. The CS was presented for 6 seconds, and CS+ presentations co-terminated with a brief electric shock (US). Therefore, the US was presented 2s after the peak SCR was scored, which was enough time to ensure that the US could not have influenced SCRs to the CS+ and the CS-. In the revised version, we also performed a correlation between SCR and fMRI responses when only including the non-reinforced CS+ trials. Results were almost identical to the main analysis including reinforced trials (see Appendix 7)"Comment: If the authors indeed employed a TTP (Trough To Peak) approach, the 1-4s post-CS refer to the onset of the SCR, not the peak. This approach is uncommon and potentially problematic as they may miss the true-peak which may occur later than 4s post-CS (see e.g. Boucsein 2012, Psychophysiolog). This needs clarification.

We have clarified our method of analysis using TTP (p. 14 rows 22-28 and p. 15 rows 1-9) and added this as a potential limitation in the manuscript (p. 6 rows 21-29). In addition, we have now revisited our SCR analysis using a different software package (PsPM). The specifics of this analysis and the results are available in Appendix 7 and 8.

On p. 14, rows 22-28, and p. 15, rows 1-9, we write: “Skin conductance was recorded with the MP-150 BIOPAC system (BIOPAC Systems, Goleta, CA). Radio-translucent disposable dry electrodes (EL509, BIOPAC Systems, Goleta, CA) were coated with isotonic gel (GEL101, BIOPAC Systems, Goleta, CA) and placed on the palmar surface of the participant’s left hand. The signal was high-pass (0.05Hz) filtered using the built-in BIOPAC hardware Butterworth filter. SCRs were scored using Ledalab software package (v 3.4.9) (Benedek and Kaernbach, 2010) implemented in Matlab 2020a (Mathworks, Inc, Natick, MA). Minimum response threshold was set to 0.01 µS. After filtering and before analysis, the SCR signal was down-sampled from 2000Hz to 200Hz (factor mean). SCR was analyzed using standard peak score (through-to-peak, TTP.AmpSum) 1-4 seconds after CS onset for each participant. To check that whole brain SCR correlations were not dependent on the choice of peak scoring window, we also analyzed SCR with a window of 1-5 seconds after CS onset. We also scored SCR using a software package called PsPM (Bach and Friston, 2013), which uses a model-based approach in estimating SCR (see Appendix 8 for details). We performed these variants of SCR scoring as part of a sensitivity analysis to ensure that correlation results between SCR and brain activity were not dependent on the choice of SCR scoring method.”

On p. 6 rows 21-29 we write: “In addition, we also repeated our analysis using an extended SCR response window as well as controlling for shock expectancy and genetic influence, again with similar results (see Appendix 7). Finally, as pointed out by a reviewer, the Ledalab software package (v 3.4.9; Benedek and Kaernbach, 2010) presently used for SCR scoring has been shown to yield no better and sometimes worse results than standard peak scoring (Bach, 2014). For this reason, we also repeated our whole brain analysis using the PsPM software package (v 5.1.1) (Bach and Friston, 2013). This analysis once again implicated the same set of regions See Appendix 7 for results and Appendix 8 for additional information on how this analysis was conducted (complementary analysis).”

"The authors seem to use a 6 s-SOA delay fear conditioning paradigm. SCR scoring was done with ledalab, using the "maximum phasic driver amplitude 1-4 seconds after CS presentation for each participant". The potential problem is the peak detection window. First, can the authors clarify whether the peak window is 1-4 s after CS onset or after CS offset? The time window is after CS onset, not offset. We have clarified this in the methods section under SCR: 'SCR was analyzed using standard peak score (through-to-peak) 1-4 seconds after CS onset for each participant' (p. 11, row 2-3)"Comment: More information is required. What kind of settings were chosen in Ledalab? What did they use "CDA. Phasicmax" or "TTP. Ampsum" for instance (or yet another option)? The information provided is too little to understand what the authors did.

The options specified for the Ledalab batch run were 'open', 'biotrace', 'downsample', 10, 'analyze','CDA', 'optimize',4. We extracted values within the response-window using TTP.AmpSum information. We updated information regarding the Ledalab analysis on p. 14, rows 22-28, and p. 15, rows 1-9.

"Second, do they analyse only non-reinforced trials or also reinforced trials? Both non-reinforced and reinforced trials were analyzed together as SCR was scored prior to US delivery. To ensure that SCR correlation with fMRI responses was equivalent for non-reinforced trials as for all trials, we analyzed these 8 trials separately, as stated earlier. Results were very similar as for all trials. We refer to the new Table 2 in our revised manuscript for statistics."Comment: I was unable to locate Table 2. Please provide Table 2 or correct the table reference.

We apologize for this mistake. We have moved the table to the appendix and have updated the reference to the table on p. 6 row 23 (Appendix 7).

"What Ledalab calls the "driver" is a peripheral neural impulse at some unspecified place in the peripheral autonomic system. As can be seen in figure 5a in Benedek and Kaernbach (2010) where SCR was elicited by external events, this "driver" peaks around 2 s after an external event. So, if the US (or US omission) elicits an SCR, then the estimated "driver" will peak 2 s after CS offset and would be included in a 1-4 s window after CS offset. If, on the other hand, there was a gradual increase in SCR > 2 s into the CS, then the driver would peak > 4 s after CS onset and would not be included in a 1-4 s window after CS onset. o In sum, the authors need to better work out and explain their peak scoring windows. They should also compare reinforced and non-reinforced CS+ trials, to rule out any bias in their analysis. Given that Ledalab yields no better results than standard peak scoring, and sometimes worse results (Bach 2014 Biological Psychology), they may want to consider using a standard peak-scoring analysis or similar strategy. (I note that the standard procedures implemented in PsPM – Bach et al. 2020 Beh Res Therapy – are not optimized for this 6-s SOA, even though there is an option that makes the models suitable for this case as well.) We thank the reviewers for this insightful comment. We have described the methodology in a more precise language. After revisiting our analysis, we noted that we had used a standard peak to through method with a time window of 1-4s post-CS onset. See our comment above to Q9."Comment: While I appreciate the revisions, this is still not clear in the revised manuscript. Please provide a coherent and adequately complete description.

We have now rewritten parts of the method with regards to how we scored SCR data to increase clarity. In the revised manuscript we have also re-analyzed our SCR data using PsPM for comparison. See p. 14, rows 22-28, and p. 15, rows 1-9. As well as Appendix 7 and 8 for updated results and methods.

"Insufficient rationale for analysis: "…if neural correlates to differential SCR were driven more by CS+ or CS-" – What is the motivation for these analyses?Individual differences in SCR difference scores could be associated both with individual differences in SCR to the CS+ and the CS-. Therefore, we wanted to check that SCR to the CS+, and not the CS-, was the reason for the observed correlation between SCR difference scores and fMRI contrast values. We have added this rationale in the results (p 6, row 23)"Comment: I was unable to locate the rationale on page 6, row 23.

We apologize for this erroneous reference. The correct reference is currently p. 9 rows 24-26.

“Individual differences in SCR difference scores could be associated with individual differences in SCR to both the CS+ and the CS-. Therefore, we wanted to test whether SCR to the CS+, and not the CS-, was the reason for the observed correlation between SCR difference scores and eigenvariates.”

"At points, the authors are insufficiently precise and nuanced in their description of prior work o For instance, please indicate the direction of published findings, rather than just reporting that there was "an association" or "altered responding". We have made changes throughout the manuscript to indicate the direction of associations between SCR and fMRI responses. We have avoided terms like "an association" or "altered responding" to more precisely indicate the directions of findings."Comment: I was unable to locate the respective changes made (as they were not referenced here and the edits in the manuscript were not fully transparent) and found the reporting oftentimes still too superficial.

We have attempted to give a more nuanced description of prior work. See the changes on p 2 rows 8-39 that we have also referenced above:

“Previous studies of the neural correlates of individual differences in conditioned SCR, generally defined as the difference in average SCR score between CS+ and CS- presentations during acquisition (see Lonsdorf and Merz, 2017, for a discussion of definitions), have focused on either one or a few brain regions, using region of interest analyses. Many of these studies have found positive correlations with neural responses in the amygdala (Labar, Gatenby, Gore, LeDoux and Phelps, 1998; Phelps, Delgado, Nearing and LeDoux et al. 2004; Dunsmoor, Prince, Murty, Kragel and Labar, 2011; Petrovic, Kalisch, Pessiglione, Singer and Dolan, 2008; MacNamara et al., 2015; Marin et al., 2019). Neuroimaging studies of within-subject variation in conditioned SCR have also generally found positive correlations to amygdala responses (Cheng, Knight, Smith, Stein and Helmstetter, 2003; Knight, Nguyen, and Bandettini, 2005; Cheng, Knight, Smith and Helmstetter, 2006), although exceptions exist (Sjouwerman, Sharfenort and Lonsdorf, 2020; Savage et al., 2021). The findings from studies that report a positive relationship between SCR and amygdala activity are in line with the general understanding of fear conditioning from animal models, where a neural circuitry centered on the amygdala is responsible for the acquisition of conditioned fear responses (LeDoux 2000; Davis 2000). They also complement those human lesion studies demonstrating either diminished or absent conditioned SCR following amygdala damage (Labar et al. 1995; Bechara et al., 1995), although not all studies have found such an effect (Åhs et al., 2010; for a review see Ojala and Bach, 2020). Further, the involvement of the amygdala in human fear conditioning has been questioned based on the results of a meta-analysis of fMRI studies investigating fear conditioning (Fullana et al., 2016) and based on studies showing unexpected, increased amygdala responses to the CS- compared to the CS+ (see e.g. Visser et al. 2021). Such results could arise from distributed representations of the CS+ and CS- in the amygdala (Bach, Weiskopf and Dolan, 2011; Reijmers, Perkins, Matsuo and Mayford 2007) or from a need for larger sample sizes to detect differential responses in the amygdala. Speaking to the latter idea, two independent studies, each including hundreds of participants, have recently reported increased CS+, relative to CS-, activation in the amygdala (Kastrati et al., 2022; Wen et al., 2022). Amygdala activation to the CS+ was primarily detected during the first trials of acquisition, whereas CS- activity was larger in the end of acquisition (Wen et al., 2022). The results of these two large and independent neuroimaging studies, together with the fairly consistent findings of correlated individual differences in conditioned SCR and amygdala activation (Labar, Gatenby, Gore, LeDoux and Phelps, 1998; Phelps, Delgado, Nearing and LeDoux et al. 2004; Dunsmoor, Prince,Murty, Kragel and Labar, 2011; Petrovic, Kalisch, Pessiglione, Singer and Dolan, 2008; MacNamara et al., 2015; Marin et al., 2019), support the hypothesis that amygdala activation should be positively correlated with SCR.”

See also the following changes in the discussion, p. 11 rows 38-52 and p. 12 rows 1-6, where we elaborate on the association between discriminative fear learning and anxiety disorders:

“As anxiety disorders have also been associated with altered discriminative fear learning (Nees, Heinrich and Flor 2015), and as our results indicate that individual differences in discriminative fear learning covary with midcingulo-insular activity, our results are consistent with this anxiety model. Specifically, in reviewing the relationship between discriminative fear learning and anxiety disorders, Nees, Heinrich and Flor (2015) note that while discriminative fear learning does not appear to be impaired across all anxiety disorders and across all stimuli types (see also Duits et al., 2015, for a meta-analysis confirming this observation), studies comparing anxiety disorder patients to controls with regard to disorder-specific stimuli have found increased discriminatory fear learning in patients (in specific phobia, see Schweckendiek et al. 2011; in social phobia, see Lissek et al. 2008; in PTSD, see Wessa and Flor 2007), suggesting that discriminatory fear learning may be a mechanism underlying these disorders. The present finding, that individual differences in discriminatory fear learning covary with differences in midcingulo-insular activity, suggests that alterations in such neural activity may also be contributing to, or resulting from, anxiety, consistent with the anxiety model proposed by Menon (2011). However, it should be noted that the present study only used SCR as an outcome measure of discriminative fear learning while the previously cited studies considered other outcome measures as well (e.g. fear potentiated startle in Lissek et al. 2008; subjective ratings of valence and arousal in Wessa and Flor, 2007), thus somewhat limiting the generalizability of our findings (for further details regarding methodological differences and similarities we refer to the review by Nees, Heinrich and Flor, 2015). We recommend that future studies continue exploring the midcingulo-insular network and its relationship to fear and anxiety disorders.”

"The authors used 4 different trial sequences. Can they provide information on which CS+ trial was the first reinforced trial in these different sequences? The reason I am asking this is that if the first 5 CS+ presentations in sequence#1 were not reinforced but already the first one was reinforced in sequence #2 this would likely lead to differences in learning speed and ultimately average CS discrimination which may impact on the results. Are individual differences in discrimination related to trial sequences? In all sequences, the first CS+ presentation following the 4 CS+ habituation trials was always reinforced. The sequences differed in whether the CS- or CS+ started the acquisition phase. If the reinforced CS+ is always the first trial in the acquisition phase, the CS- trial following the shock will be elevated due to sensitization. This was why the presentation order was counterbalanced. Although it is possible that trial sequences may be related to discrimination, if this is the case, we still show that the individual variation in SCR correlates with fMRI responses irrespective of trialorder."Comment: Please add this useful information to the manuscript.

We thank the reviewer for this suggestion. We have added this information to the revised manuscript on p 16 rows 7-10.

"Please clarify in the text whether the amygdala was significantly activated in the whole-brain CS+ vs CS- contrast, as this will be useful for other investigators and future meta-analyses. We now write in the Results section: "We found no differences in neural responses to the CS+ compared to the CS- during habituation. During acquisition, the pattern of activation to the CS+ relative to the CS- was very similar to the pattern reported in the meta-analysis by Fullana et al. (2016) and included large parts of the striatum, the insula, midline areas of the cingulum, lateral temporal cortex, parietal cortex and the supplementary motor areas. Of note, the whole-brain analysis also revealed greater activation to the CS+ than to the CS- in the bilateral amygdala."Comment: I was confused about this section (page 3 line 45 to end of page) as the authors compare their results to those of Fullana who looked at the CS+/CS- contrast in fMRI but not at a correlation with SCRs. I think this is also done in other sections of the manuscript. It needs to be made clear in the MS that Fullana did not investigate brain-behavior associations (i.e. neural correlates of differential SCRs and how the results relate to each other).

We have made it clearer that the meta-analysis by Fullana et al. did not investigate correlations with SCR throughout the manuscript. We mention this study when we suggest that the correlations to SCR may include other brain areas than the ones investigated in the previous correlation studies. We hope it is clear that Fullana et al. investigated the neural activation related to the CS+ and CS-. We write on p. 5, rows 2-4: “Fear conditioning is known to activate a large set of cortical, subcortical, and brainstem areas other than the ROIs that have so far been investigated for their correlation with SCR (Fullana et al., 2016).”

"Discussion 41 o Authors note that individual differences in SCRs are stable and provide 3 references for this. They may want to double-check if these references really show demonstrate the stability of individual differences in CS discrimination. If I am not mistaken, neither Fredriksson (1993) nor Zeidan (2012) report stability measures for CS discrimination per se (but only for CS+ and CS- individually). This is a good point. We now refer to Fredrikson (1993) and Zeidan (2012) in terms that they have shown that SCR during fear conditioning is relatively reproducible. We do not refer to CS differences here. (First sentence in the discussion, p. 7, row 5)"Comment: Technically, what was studied by Fredrikssion and Zeidan was reliability, not reproducibility (see above for a definition of what reproducibility refers to). I suggest being more precise here. Also, I do not think it becomes clear from the revision that the findings by Zeidan, Torrents-Rodas and Fredriksson do not refer to CS discrimination (or do they?). On the contrary, from the wording the authors have chosen, I would infer that this is about CS discrimination. As this work is mainly about CS discrimination this is important. I refer the authors to other work that also investigated fMRI test-retest reliability and/or test-retest reliability for CS discrimination and once more would appreciate (again) more precision in reporting.PublishedRidderbusch, I. C., Wroblewski, A., Yang, Y., Richter, J., Hollandt, M., Hamm, A. O.,.…Straube, B. (2021). Neural adaptation of cingulate and insular activity during delayed fear extinction: A replicable pattern across assessment sites and repeated measurements.NeuroImage, 237, 118157. https://doi.org/10.1016/j.neuroimage.2021.118157Pre-printsSamuel E Cooper, Joseph E Dunsmoor, Kathleen Koval, Emma Pino, Shari Steinman, Test-Retest Reliability of Human Threat Conditioning and Generalization , PsyArXiv, https://psyarxiv.com/84uqz/Maren Klingelhöfer-Jens, Mana R. Ehlers, Manuel Kuhn, Vincent Keyaniyan, Tina B. Lonsdorf. Robust group- but limited individual-level (longitudinal) reliability and insights into cross-phases response prediction of conditioned fear doi: https://doi.org/10.1101/2022.03.15.484434

We thank the reviewer for noticing that the reliability studies of SCR refer to responses to either the CS+ or CS-, but not the difference between CS+ and CS-. We have now removed these references along with the statement that individual differences in CS discrimination are reliable (p. 10 rows 9-11). As shown by the preprints referenced by the reviewer the findings regarding individual-level reliability in CS discrimination measured by SCR and fMRI BOLD are mixed (Cooper et al.; KlingelhögerJens et al.; Ridderbusch et al., 2021). We have therefore chosen to instead mention this as a limitation later on in the discussion (P. 12 rows 28-41).

“Finally, it should be noted that only a few studies to date have examined the longitudinal (test-retest) reliability of individual differences in discriminatory fear learning measured by SCR (Cooper, Dunsmoor, Koval, Pino and Steinman, 2022, Preprint; Klingelhöfer-Jens, Ehlers, Khun, Keyaniyan and Lonsdorf, 2022, Preprint). While one study found evidence of fair withinperson stability in 51 participants across a 9-day period (Cooper et al., 2022, Preprint), another study found evidence of poor individual-level reliability in 120 participants across a 6month period (Klingelhöfel-Jens et al., 2022, Preprint). Similarly, previous studies examining the individual-level reliability of fMRI BOLD responding during fear conditioning have reported low to moderate reliability (Ridderbusch et al., 2021; Klingelhöfel-Jens et al., 2022, Preprint). Taken together, this means that it is currently unclear to what extent our results reflect stable individual traits, as opposed to participants’ particular states at the time of measurement, which limits the interpretation of our findings. In line with Klingehöfel-Jens et al. (2022, Preprint) we encourage future research on individual differences in fear conditioning to explore new ways of improving the reliability of measurement.”